# Atomic-resolution mapping of transcription factor-DNA interactions by femtosecond laser crosslinking and mass spectrometry

Alexander Reim [1], Roland Ackermann [2], Jofre Font-Mateu[3], Robert Kammel[2], Miguel Beato[3,4], Stefan Nolte [2,5], Matthias Mann [1], Christoph Russmann [6,7✉] & Michael Wierer [1✉]

Transcription factors (TFs) regulate target genes by specific interactions with DNA sequences. Detecting and understanding these interactions at the molecular level is of fundamental importance in biological and clinical contexts. Crosslinking mass spectrometry is a powerful tool to assist the structure prediction of protein complexes but has been limited to the study of protein-protein and protein-RNA interactions. Here, we present a femtosecond laser-induced crosslinking mass spectrometry (fliX-MS) workflow, which allows the mapping of protein-DNA contacts at single nucleotide and up to single amino acid resolution. Applied to recombinant histone octamers, NF1, and TBP in complex with DNA, our method is highly specific for the mapping of DNA binding domains. Identified crosslinks are in close agreement with previous biochemical data on DNA binding and mostly fit known complex structures. Applying fliX-MS to cells identifies several bona fide crosslinks on DNA binding domains, paving the way for future large scale ex vivo experiments.

[1] Department of Proteomics and Signal Transduction, Max-Planck Institute of Biochemistry, Am Klopferspitz 18, 82152 Martinsried, Germany. [2] Institute of Applied Physics, Abbe Center of Photonics, Friedrich-Schiller-Universität Jena, Albert-Einstein-Straße 15, 07745 Jena, Germany. [3] Centre for Genomic Regulation (CRG), The Barcelona Institute of Science and Technology, Dr. Aiguader 88, 08003 Barcelona, Spain. [4] University Pompeu Fabra (UPF), 08002 Barcelona, Spain. [5] Fraunhofer Institute for Applied Optics and Engineering (IOF), Albert-Einstein-Straße 7, 07745 Jena, Germany. [6] University of Applied Sciences and Arts Hildesheim/Holzminden/Goettingen (HAWK), Von-Ossietzky-Straße 99, 37085 Göttingen, Germany. [7] Brigham and Women's Hospital, Harvard Medical School, 75 Francis Street, Boston, MA 02115, USA. ✉email: christoph.russmann@hawk.de; wierer@biochem.mpg.de

Transcription factors (TFs) are key players in the regulation of gene expression and control a multitude of cellular functions, including differentiation, maintenance of cellular identity, cell homeostasis, as well as highly cell specific functions such as immune response[1]. Due to their pivotal role in cellular signaling, mutations of TFs are often linked to human diseases[2–4].

TFs exert their gene regulatory function through the recognition of specific DNA-binding elements in spatial vicinity of target genes and by the recruitment of coregulators, which may have transcriptional activating or repressing functions. DNA binding is mediated by specific DNA-binding domains (DBDs). Evolution gave rise to various different classes, including zinc finger, HMG-box, leucine zipper, helix-turn-helix, and helix-loop-helix domains[1]. Most DBDs of known and putative TFs are identified and classified by sequence homology to a previously characterized DBD[5] and large-scale studies verified the DNA-binding specificity of several hundred individual domains[6,7]. Nevertheless, for several DNA-binding proteins the DBD is unknown, due to the lack of homology with classical domains. Even for domains that have been proven to bind DNA in a stand-alone context, it is not certain that the domain will have the same functionality in the full-length protein.

The molecular mechanism by which TFs bind to DNA can be elucidated by cocrystallization of protein–DNA complexes, which provides insight into the amino acids that are in closest vicinity to the DNA and therefore most likely involved in DNA binding[8,9]. NMR spectroscopy has been used to gain similar information[10]. Furthermore, the composition and stoichiometry of large protein–DNA complexes can be disentangled using high-resolution electron microscopy (EM)[11]. While all those methods allow to study protein–DNA complexes in great detail, for many TFs they are very time consuming or not feasible at all. In addition, especially for crystallization, they reflect a frozen state, which can be different from the dynamic binding behavior of TFs to DNA in solution.

With the advances in mass spectrometry (MS) over the past decade[12], cross-linking MS (XL-MS) has become a viable complementary method to study the structure of protein complexes. The use of chemical crosslinkers allowed the analysis of stoichiometry and spatial arrangement of proteins organized into large complexes (reviewed in ref. [13]). More recently, XL-MS has also entered the field of protein–RNA interactions. Here, ultraviolet (UV) irradiation can create "zero-length" cross-links in the native state of a protein–RNA complex, meaning the direct covalent attachment of an amino acid to a nucleotide. Pioneering studies applied UV irradiation and MS analysis to identify RNA-binding proteins on a system-wide scale in yeast and mammalian cells[14–16]. Improvements in bioinformatic tools further allowed the localization of RNA-protein cross-links at the level of single amino acids[17], providing complementary information about RNA-binding domains.

Despite these developments in applying UV XL-MS to study protein interaction with RNA, the technology has not been applicable for protein–DNA interactions so far. This is largely due to the fact that double-stranded oligonucleotides are about an order of magnitude less efficiently cross-linked by UV than single-stranded oligonucleotides[18]. Yet, over the last three decades a small number of studies have shown that the efficiency of protein–DNA cross-linking can be increased by using UV lasers[19–24]. For a given total energy, the efficiency of protein–DNA cross-linking was shown to largely depend on the length of the laser pulses. Highest cross-linking efficiency can be reached with an ultrafast femtosecond laser, providing 30 times higher efficiency than a nanosecond laser[20].

To map protein–DNA interaction in a highly specific manner, we here present a pipeline for femtosecond UV-laser-induced cross-linking combined with high-resolution MS (fliX-MS). Our workflow is capable of mapping protein–DNA interactions of in vitro assembled nucleosomes as well as in vitro and ex vivo TF–DNA interactions. Our method successfully confirms protein–DNA binding sites predicted by structural studies, and provides insights into the extent of flexibility within DBDs.

## Results

**A fliX-MS pipeline to map protein–DNA interactions**. UV-laser cross-linking with ultrafast pulses can cross-link TFs and DNA with high efficiency[20]. Here, we developed a pipeline, which combines that technology with a high-resolution MS methodology in order to map DNA–protein interactions on amino acid level (Fig. 1). To this end, we used a femtosecond fiber laser at 515 nm, and further doubled its wavelength to 258 nm with a beta barium borate (BBO) crystal (Fig. 1a). Its frequency was 0.5 MHz and pulse duration about 500 fs. The laser beam was adjusted to 2.5 mm ($e^{-2}$), in order to match the inner diameter of a 1.5 ml Eppendorf tube containing the sample. Following UV irradiation, we denatured protein–DNA complexes, cut the DNA to mono or short oligonucleotide size using a mix of three different nucleases, and digested proteins to peptides with trypsin and Lys-C (Fig. 1b). We then separated peptides from free DNA with StageTips loaded with C18 material[25], enriched peptide–DNA cross-links using titanium dioxide ($TiO_2$) coated beads, and analyzed them by high-resolution MS (see "Methods"). Peptide–DNA cross-links were searched in MS data using the RNP(xl) software, which was originally developed for the identification of peptide–RNA cross-links[17] (Fig. 1c). Processing nonirradiated control samples in parallel allowed us to subtract any spectra that were not UV cross-linking specific, massively reducing the search space. To improve detection of true DNA cross-links, we further manually validated and annotated all cross-linked peptide fragmentation spectra, considering y-, b-, and a-ion series, as well as internal fragment ions.

**Optimization of cross-linking conditions**. To maximize the cross-linking rate and therefore the identification of protein–DNA cross-links, we first optimized the femtosecond UV-laser parameters. UV-dependent DNA cross-linking is a two-photon process and depends on both intensity and pulse length[20]. As the pulse length is determined by the laser setup, we tested different pulse energies, as well as increasing amounts of total energy.

We used a recombinant TF—porcine nuclear factor 1/C (NF1) —and let it bind to a biotinylated oligonucleotide containing its specific DNA-consensus binding site or a mutated version of it (Fig. 2a). As the binding was much stronger for the wild-type binding site, compared with its mutant counterpart, we concluded that the protein–DNA interaction was functional. The minor binding to the mutant consensus site can be explained by the ability of NF1 to bind DNA also in unspecific manner[26]. Next, we UV-irradiated the NF1–DNA complex with a pulse energy of 7 nJ and increasing amounts of total energy followed by western blotting and detection of protein–DNA cross-links using a streptavidin–HRP conjugate (Fig. 2b). There was a direct relationship between total energy and cross-linking yield at the beginning of the curve and only a minor increase of cross-linked species from 350 mJ onward. With higher total energy, we also observed protein–protein cross-links bound to biotin-DNA, reflected in an increasing signal in the higher molecular weight range (Supplementary Fig. 1a).

**Fig. 1 Schematic workflow of the fliX-MS pipeline. a** A pulsed laser beam was generated using a femtosecond fiber laser with 515 nm wavelength, repetition rate of 0.5 MHz, and pulse duration of 500 fs. The wavelength was doubled to 258 nm by second harmonic generation (SHG) over a beta barium borate (BBO) crystal and the laser beam adjusted to fit the inner diameter of a regular 1.5 ml Eppendorf tube. **b** Protein–DNA complexes were irradiated or left untreated as control. Samples were denatured, DNA digested to mono/short oligonucleotides by a mix of Mnase, DNase I, and Benzonase, and proteins digested to peptides by trypsin and Lys-C. Peptides and peptide–nucleotide cross-links were separated from free DNA on C18 StageTips[25], and cross-links subsequently enriched with TiO₂ beads. **c** Peptides were measured by LC-MS/MS and data analyzed with the RNP(xl) software package implemented in the proteome discoverer software[50] followed by manual annotation of candidate spectra.

To determine the optimal pulse energy, we next irradiated the TF–DNA complex with increasing pulse energies, keeping the total energy at 1 J (Fig. 2c, Supplementary Fig. 1b). Maximum cross-linking efficiency occurred at about 40 nJ pulse energy, whereas it strongly decreased at both lower and higher pulse energies. While the lower cross-linking efficiency with less pulse energy can be explained by a minimum energy requirement for the two-photon processes to take place, the reduction at higher pulse rates is either due to saturation effects or DNA damage. We conclude that a maximal energy of 50 nJ per pulse is sufficient to cross-link protein–DNA complexes, and an increase of pulse energy does not enhance the process.

To investigate whether the formed protein–DNA cross-links reflected functional TF–DNA interactions, we repeated the titration of the total pulse energy with the optimal pulse energy of 50 nJ for NF1 bound to a DNA oligo containing either its wild-type consensus binding sequence or a mutated form of it (Fig. 2d, Supplementary Fig. 1c). Western Blot analysis of the biotin-DNA complex revealed that protein–DNA cross-linking was specific for the wild-type sequence. Notably, this was also the case for the higher molecular weight fraction, indicating that protein–protein cross-linking does not affect DNA-binding specificity, even at a total energy of 1.25 J.

To quantify the cross-link efficiency, we irradiated NF1–DNA complex (pulse energy of 7 nJ and a total energy of 350 mJ) and probed the western blot with an antibody directed against the His-tag of NF1 (Fig. 2e, Supplementary Fig. 1d). We observed a shifted double band at 60–65 kDa, which disappeared when digesting the sample with either DNase I or proteinase K suggesting that the signal is derived from the NF1 bound to single- and double-stranded DNA. Reblotting the stripped membrane with the streptavidin–HRP conjugate recognizing biotinylated DNA confirmed this observation. Quantification of the mono-NF1–DNA cross-links revealed a cross-linking efficiency of 7.5%. Taking into account also the high-molecular weight population and extrapolating from the cross-linking efficiency of mono-NF1–DNA and the intensities of the 65, 130, and 185 kDa bands in the DNA-biotin blot, we estimate a cross-linking efficiency of 14% under these energy conditions (Supplementary Fig. 1d).

To validate the observations with another TF–DNA complex, we UV-irradiated recombinant TATA-box binding protein (TBP) bound to an oligo containing either the wild-type TATAA sequence or a single point mutant of it (TGTAA), known to decrease TBP binding by 49%[27] (Fig. 2f). As expected, we observed a stronger signal for the TBP–TATAA complex compared with the TBP–TGTAA, which disappeared with either DNase I or proteinase K treatment indicating that fliX-MS works effectively also for TBP. Of note, the difference in the cross-link efficiency for the two sequences was also visible in the high-molecular weight fraction, corresponding to multiple copies of TBP bound to DNA (Supplementary Fig. 1e).

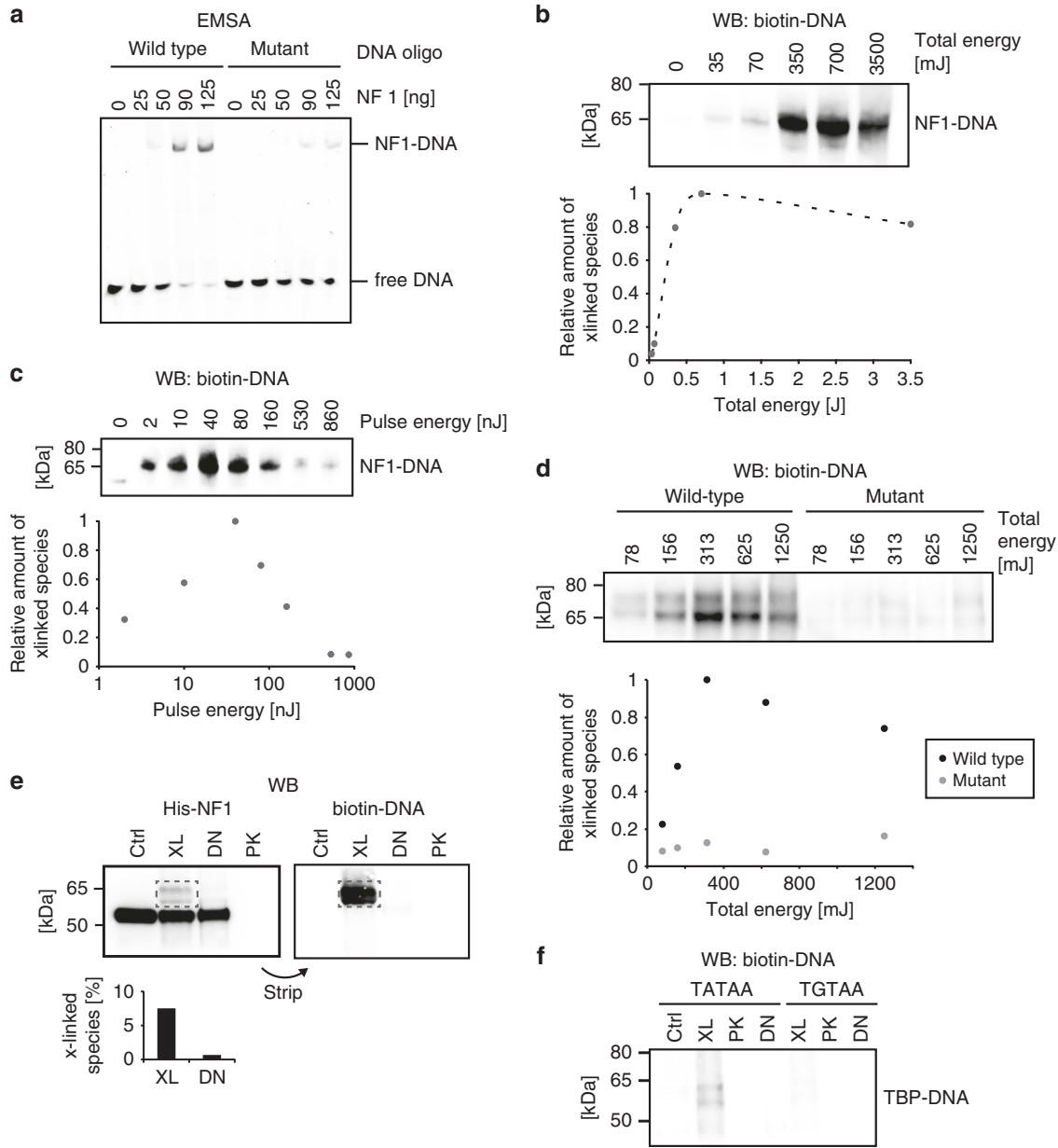

**Fig. 2 Assessment of cross-linking efficiencies. a** Electrophoretic mobility shift assay (EMSA) with increasing amount of NF1 bound to a DNA oligomer harboring its consensus site, or a mutated version of it. The molar ratios of protein to DNA were 2.1:1 (25 ng), 4.4:1 (50 ng), 8:1 (90 ng), and 11:1 (125 ng). The NF1–DNA complex was separated from free DNA by nondenaturing gel electrophoresis and visualized by SYBR Green staining. **b** NF1–DNA (5'-biotinylated) complex was irradiated with increasing total energy and constant pulse energy of 7 nJ. Samples were separated by denaturing gel electrophoresis, protein–DNA complexes transferred to a nitrocellulose membrane, and biotinylated DNA visualized by probing with an HRP-coupled streptavidin conjugate. Intensities of the cross-linked protein–DNA bands (x-linked species) were quantified and plotted relative to the most intense band at 700 mJ. **c** NF1–DNA (5'-biotinylated) complex was cross-linked applying increasing pulse energies, and a constant total energy of 1 J. Cross-linked protein–DNA complexes were detected as in **b**. Band intensities were plotted relative to the most intense band at a pulse energy of 40 nJ. **d** NF1 bound to a DNA oligo harboring its consensus site or a mutated version of it was irradiated with increasing total energy and constant pulse energy of 50 nJ: cross-linking depended on a functional protein–DNA interaction. **e** NF1–DNA complex was cross-linked with a pulse energy of 7 nJ and 350 mJ total energy (XL) or left untreated (Ctrl). Cross-linked samples were further optionally treated with DNase I (DN) or proteinase K (PK) and loaded on a SDS-PAGE followed by western blotting. After detection of His-NF1 using an anti-His antibody, the membrane was stripped and reprobed with an HRP-coupled streptavidin conjugate to detect biotin-labeled DNA. The percentage of cross-linked protein–DNA complexes (x-linked species) was calculated as the intensity of the cross-linked band (dashed rectangle) divided by the sum of intensities of all bands observed in the cross-linked sample. **f** TBP bound to DNA oligos containing either a wild-type (TATAA) or point-mutated (TGTAA) consensus motif were UV irradiated (pulse energy 50 nJ, total energy 1.25 J) and biotin-DNA detected by western blot. Full-scale versions of all blots are depicted in Supplementary Fig. 1.

**a**

| DCP id | Histone | Peptide sequence | Crosslinked nucleotide | Charge | m/z | Δm [ppm] |
|---|---|---|---|---|---|---|
| 1 | H4 | [21]KVLRDNIQGITKPAIR[36] | ACC**G**-$H_3PO_4$ | 3 | 1002.4468 | 6.2 |
| 2 | H4 | [22]VLRD**N**IQGITKPAIR[36] | C**T**-$H_2O$ | 3 | 777.3659 | 9.5 |
| 3 | H3.1 | [29]**S**APATGGVK[37] | **C**-$HPO_3$ | 2 | 507.7652 | 1.6 |
| 4 | H3.1 | [74]**EIAQ**DFK[80] | A**T**-$HPO_3$ | 2 | 703.2883 | 6.5 |
| 5 | H3.1 | [118]V**T**IMPKDIQLAR[129] | **C**-$HPO_3$ | 3 | 543.3030 | 6.7 |
| 6 | H2B-1K | [45]VLKQ**V**HPDTGISSK[58] | AAG**T**+$HPO_3$ | 2 | 1433.5139 | 4.6 |
| 7 | H2B-1K | [45]VLKQ**V**HPDTGISSK[58] | AGG**T**+$HPO_3$ | 2 | 1441.5155 | 1.7 |
| 8 | H2B-1K | [110]HAVS**EGTKAV**TK[121] | **G**T | 2 | 961.3941 | 1.7 |
| 9 | H2A-1B | [79]IIPRHLQLAIR**N**DEELNK[96] | A**T** | 4 | 702.5960 | 8.3 |
| 10 | H2A-1B | [83]HLQLAIRNDEE**LNK**[96] | **T**-$HPO_3$ | 3 | 660.0110 | 10.0 |
| 11 | H2A-1B | [83]HLQLAIR[89] | AA**T**-$HPO_3$ | 3 | 573.5804 | 2.6 |
| 12 | H2A-1B | [90]**ND**EELNK[96] | A**C** | 2 | 741.2575 | 1.1 |

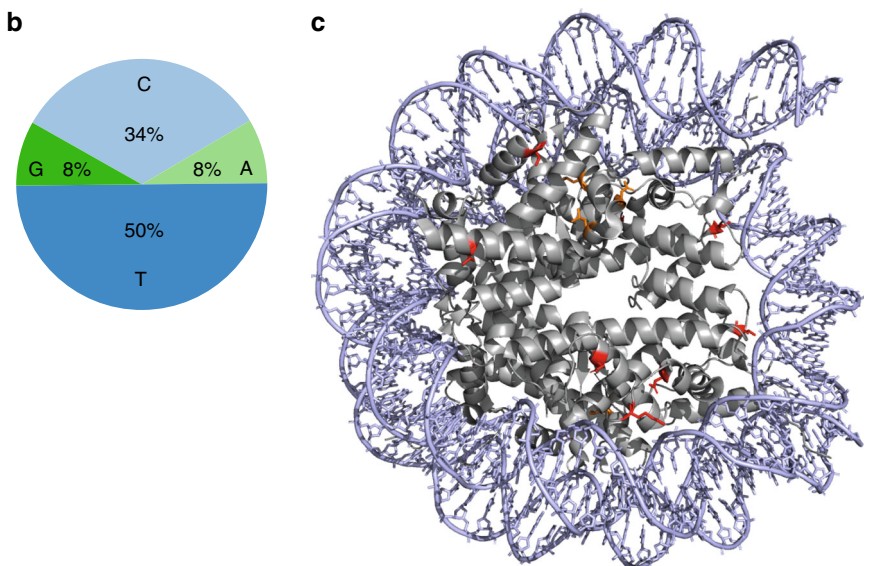

**Fig. 3 fliX-MS of in vitro assembled human nucleosomes. a** fliX-MS revealed 12 unique peptide–nucleotide cross-links. The sequences of the cross-linked peptides are shown. For cross-links that could be located on one or several amino acids, the location within the peptide is marked in red letters. The cross-linked nucleotide sequence derived from precursor mass differences (A: deoxyadenosine, C: deoxycytidine, G: deoxyguanosine, T: thymidine), charge state, mass-to-charge ratio (*m/z*), and mass error (Δ*m*) are shown. The cross-linked base is marked in red letters. **b** Base distribution among cross-links. **c** Crystal structure of the human recombinant nucleosome (PDB ID: 2CV5[8]), with cross-linked amino acids marked in red (close to DNA) and orange (distant to DNA). For cross-links with more than one potential cross-linked amino acid, the residue closest to the DNA is marked.

**Protein–DNA cross-linking of recombinant human nucleosomes.** We next applied the fliX-MS workflow to in vitro assembled human nucleosomes, as this structure involves a large number of protein–DNA contacts. This identified 12 unique peptide–nucleotide cross-links, located on seven different peptides (Fig. 3a, Supplementary Data 1). The cross-linked peptides had MS1 mass shifts corresponding to one to four nucleotides. Considering the base specific MS2 mass shifts, we were able to unambiguously call the nucleotide that was cross-linked in all of the DNA-modified peptides. Cross-links to nucleotides of pyrimidine bases represented the large majority, with six and four cross-links on thymidine and deoxycytidine, respectively. However, fliX-MS also revealed cross-links to nucleotides with purine bases, with one cross-link to deoxyadenosine and one to deoxyguanosine (Fig. 3b). This imbalance between the different base classes is likely due to their different susceptibility to the two-photon processes[28]. In any case, our results show that ultrashort laser UV pulses are capable to cross-link nucleotides of all four bases.

Cross-link-derived mass shifts in MS2 spectra also allowed the localization of the cross-link within the DNA-modified peptides. In seven cases we could pinpoint the cross-link to a single amino acid and in five other cases, we could narrow down the cross-link localization to stretches of two to six amino acids (Fig. 3a).

Comparing our results with the crystal structure of the human nucleosome[8], 8 of the 12 cross-links were in close vicinity to the DNA, with side chains of the respective amino acids pointing toward the DNA double helix (Fig. 3c). Yet, for four DNA-cross-linked peptides (DCPs 9–12, Fig. 3a, c), the distance of the closest possible cross-linked amino acid to the DNA was between 16.5 and 22.1 Å and therefore too large to be explained by a direct protein–DNA contact. As nucleosomes are known to undergo structural changes due to transient unwrapping of DNA[29,30], we hypothesized that the distant cross-links were derived from different conformational states that are not reflected in the crystal structure. In support of this notion, all cross-links that were unexpectedly far away from the DNA in the nucleosome structure, were located on the α3-helix of H2A, which is

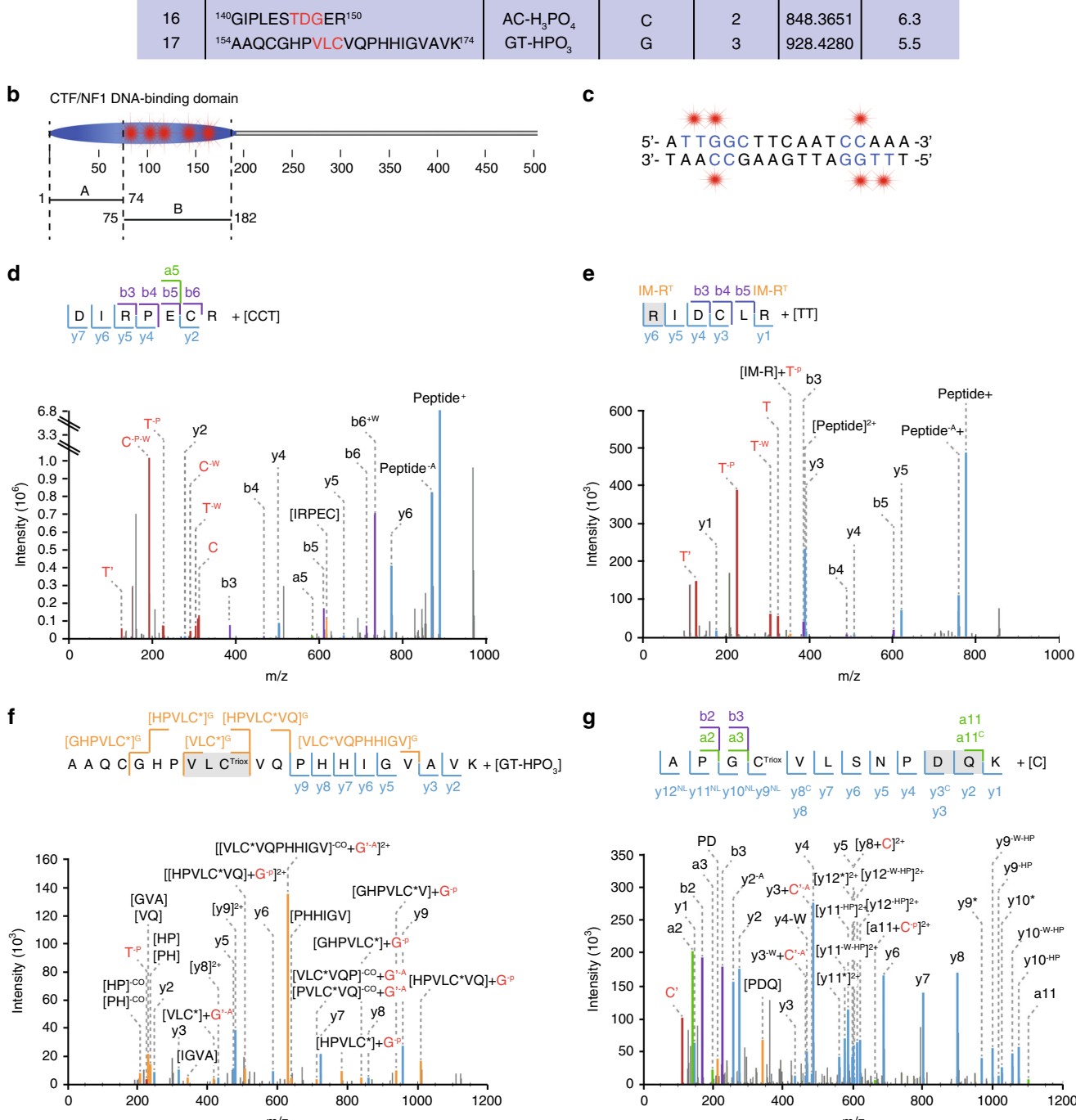

**a**

| DCP id | XL-peptide sequence | XL-nucleotide | XL-base | Charge | m/z | Δm [ppm] |
|---|---|---|---|---|---|---|
| 13 | [83]DIRPECR[89] | CCT | C | 2 | 894.7907 | 6.0 |
| 14 | [101]APGCVLSNPDQK[112] | C | C | 2 | 792.3290 | 6.4 |
| 15 | [117]RIDCLR[122] | TT | T | 2 | 701.2614 | 8.2 |
| 16 | [140]GIPLESTDGER[150] | AC-H$_3$PO$_4$ | C | 2 | 848.3651 | 6.3 |
| 17 | [154]AAQCGHPVLCVQPHHIGVAVK[74] | GT-HPO$_3$ | G | 3 | 928.4280 | 5.5 |

particularly rearranged during partial unwrapping of DNA from the nucleosome[29,30]. We therefore conclude that fliX-MS is able to detect different conformational states of a protein–DNA complex in solution.

**fliX-MS applied to the NF1–DNA complex.** Next, we enriched peptide–DNA cross-links from the NF1–DNA complex following femtosecond laser irradiation. Subjecting the cross-linked peptides to high-resolution MS, we identified five unique peptides

shifted by a mass corresponding to mono-, di-, or trinucleotides in the precursor ions (Fig. 4a). All cross-linked peptides were part of the DBD of the porcine nuclear factor 1/C (amino acids 2–195), demonstrating the structural specificity of fliX-MS (Fig. 4b). In addition, all cross-links were located on peptides between amino acids 83 and 174 indicating a specific binding region in this part of the protein. NF1 and especially its CTF/NF1–DBD are highly conserved across species (Supplementary Fig. 2). Previous experiments using truncated versions of rat NF1 showed that amino acids 75–182 were responsible for

**Fig. 4 Mapping protein–DNA interactions in the transcription factor nuclear factor 1/C. a** Overview of the identified peptide–nucleotide cross-links. The possible cross-link locations are indicated by red letters in the peptide sequence. Cross-linked (XL)-nucleotide and XL-base information derived from specific MS1 and MS2 mass shifts are specified. **b** Location of the annotated DNA-binding domain of nuclear factor 1/C (NF1) and location of the detected cross-links (red stars). A represents the unspecific DNA-binding subdomain and B the sequence-specific DNA-binding subdomain according to Dekker et al.[26]. **c** Location of the cross-links (red stars) on the palindromic consensus DNA-binding sequence of NF1. Blue letters indicate nucleotides, which fit to the NF1 consensus sequence TTGGC(N)6CC[32]. **d–g** MS2 ion series and spectra of four NF1–DNA cross-links. In the MS2 spectra, nucleotides are annotated in red, amino acids in regular letters. N' denotes the nucleobase, and N the deoxynucleotide monophosphate (with N being one of the four bases A/T/G/C). The following abbreviations describe neutral losses after MS2 fragmentation: Asterisk: neutral loss of $H_2SO_3$, $-CO$: neutral loss of carbon monoxide, $-A$: neutral loss of ammonia, $-/+W$: neutral loss or adduct of water, $-HP$: neutral loss of hydrogen peroxide, $-p$: neutral loss of $HPO_3$, $-P$: neutral loss of $H_3PO_4$. In the MS2 ion series, cross-linked fragments are depicted with the cross-linked nucleotide (A/T/G or C) in superscript. $M^{Ox}$ represents oxidated methionine and $C^{Triox}$ trioxidated cysteine (cysteic acid). The prefix IM before the respective amino acid indicates an immonium ion. The superscripted NL represents the neutral loss of sulfurous acid or hydrogen peroxide. All other symbols represent the same neutral losses as in the MS2 spectra.

sequence-specific DNA binding, while amino acids 1–78 had only nonspecific DNA-binding affinity[26]. Notably, all our cross-linked peptides located in the region responsible for sequence-specific DNA binding, highlighting the capability of fliX-MS to detect specific protein–DNA contacts (Fig. 4b, c).

For all cross-linked peptides, we defined the nucleotides that were cross-linked to the peptides making use of characteristic differences in the precursor mass. In addition, specific product ion mass shifts in the MS/MS spectra allowed us to define the exact bases that formed the cross-links (Fig. 4a, d–g). In addition to three cytosine cross-links and one thymine cross-link, one cross-link occurred to guanine, once more underscoring the potential of fliX-MS to cross-link purine bases.

The DNA contact sites of NF1 are known from DNA modification studies[31,32]. To a large extent, DNA binding is mediated by contact to the TTGG motif in the forward strand, as well as additional nucleotides in the reverse strand, which point in the same direction of the double helix (Fig. 4c). Our cross-link data covered interactions of the TTGG motif with two unique peptides (DCPs 15 and 17). In addition, we identified three cytosine cross-links, two of which were specific for the reverse strand (DCPs 13 and 16). While cytosine interactions have not been investigated previously, our data strongly suggest binding to the cytosines opposite of the TTGG sequence. Taken together, all identified cross-links fit to the defined NF1 consensus motif TTGGC(N)6CC[32].

In four out of the five DCPs, mass shifts in the MS2 spectra allowed us to locate the interactions to one, two, or three amino acids. For instance, the peptide RIDCLR cross-linked to a thymidine dinucleotide (DCP15), revealed a specific marker ion of the mass of an arginine immonium ion shifted by the mass of thymidine (Fig. 4e). As the presence of a DNA cross-link on the C-terminal arginine is unlikely due to steric interference during trypsin digest, we allocated the cross-link to R117. This residue is in close vicinity to L121/R122, which in a previous mutation study conferred DNA-binding activity of NF1[33]. On the same line, the seven amino acid long DCP13, which did not reveal a specific cross-linked amino acid (Fig. 4d), overlaps with the C88/R89 mutation site, which also significantly reduced DNA-binding affinity in the previous study.

Analysis of fragment spectra of the other cross-linked NF1 peptides provided additional technical characterization of fliX-MS. Both C104 and C163 were trioxidated to cysteic acid, likely as a result of sample preparation under nonreducing conditions[34–36] (Fig. 4f–g). In the MS2 fragmentation, the trioxidized cysteine underwent neutral loss of sulfurous acid $H_2SO_3$ (Fig. 4f–g), as has been reported previously[37]. Yet, in case of [101]APGCVLSNPDQK[112], we also observed an alternative neutral loss of 34.005 Da, which corresponds to the molecular weight of hydrogen peroxide $H_2O_2$ (Fig. 4g). Moreover, we observed multiple fragments with neutral losses of

ammonia on the guanine (Fig. 4f) and cytosine base (Fig. 4g). Such neutral losses have been reported previously for the measurement of free guanine, cytosine, and adenine per MS[38–40]. Including neutral loss of ammonia in the search for MS2 fragment ions that are characteristic for these base adducts strongly enhanced the capability of localizing DNA modifications on individual amino acids. In case of DCP14, the loss of the mononucleotide indicates a cross-link between the amino group of cytosine and the aspartate side chain, which dissociated during higher-energy collisional dissociation (HCD) fragmentation.

**Cross-linking of the TATA-box binding TF TBP**. We next applied the fliX-MS workflow to human TBP bound to the adenovirus major late promoter containing a TATA box. MS analysis of the cross-linked protein identified four cross-links on three unique peptides (Fig. 5a). As in the case of NF1, all of the TBP peptides with DNA modifications were exclusively located on the DBD of TBP (Fig. 5b).

The precursor of the peptide [255]IQNMVGSCDVK[265] was shifted by the mass of a TT-$HPO_3$ dinucleotide. Detailed analysis of the MS2 spectrum narrowed down the cross-link to either N257 or M258 (Supplementary Data 1). In the crystal structure of TBP bound to the Adml promoter[41,42], N257 is in close contact to the DNA and located between the two thymines and the two adenines of the complementary strand (Fig. 5c). The distance to either of the thymines is very short with 6.1 or 6.3 Å, respectively, thus both thymines are likely to be cross-linked to the contacting aspartic acid.

In addition, we observed an adenine cross-link to one of the amino acids G217–V220 (Fig. 5d). Based on information from the crystal structure, V220 has been mapped to interact with an adenine in the TATA box[9,42], given an extremely short distance of 3.5 Å (Fig. 5c). Hence, also this cross-link fits to the published structure with high probability. Notably, the same peptide, which contains the V220-A modification, has a second cross-link to a cytosine on A211, which in the crystal structure is located on the fourth strand of the beta sheet (Fig. 5c, d). The closest cytosine is the first nucleotide downstream of the TATAAAA sequence, on the opposite strand, with a distance of 13.4 Å. The coexistence of both cross-links on the same peptide indicates that A211 infers additional DNA binding of TBP, reaching toward a nucleotide adjacent to the TATA box.

The third TBP DCP ([178]LDLKTIALR[186]) reflected a cross-link of a cytosine to L178 (Supplementary Fig. 3a). This leucine is located between the four adenine bases and the following guanine stretch downstream of the TATA box. The closest cytosine is the same nucleotide, which we found cross-linked to A211. However, compared with the other TBP cross-links, the distance in the crystal structure to the cytosine is comparably large (17.3 Å, Supplementary Fig. 3b). One explanation to this discrepancy

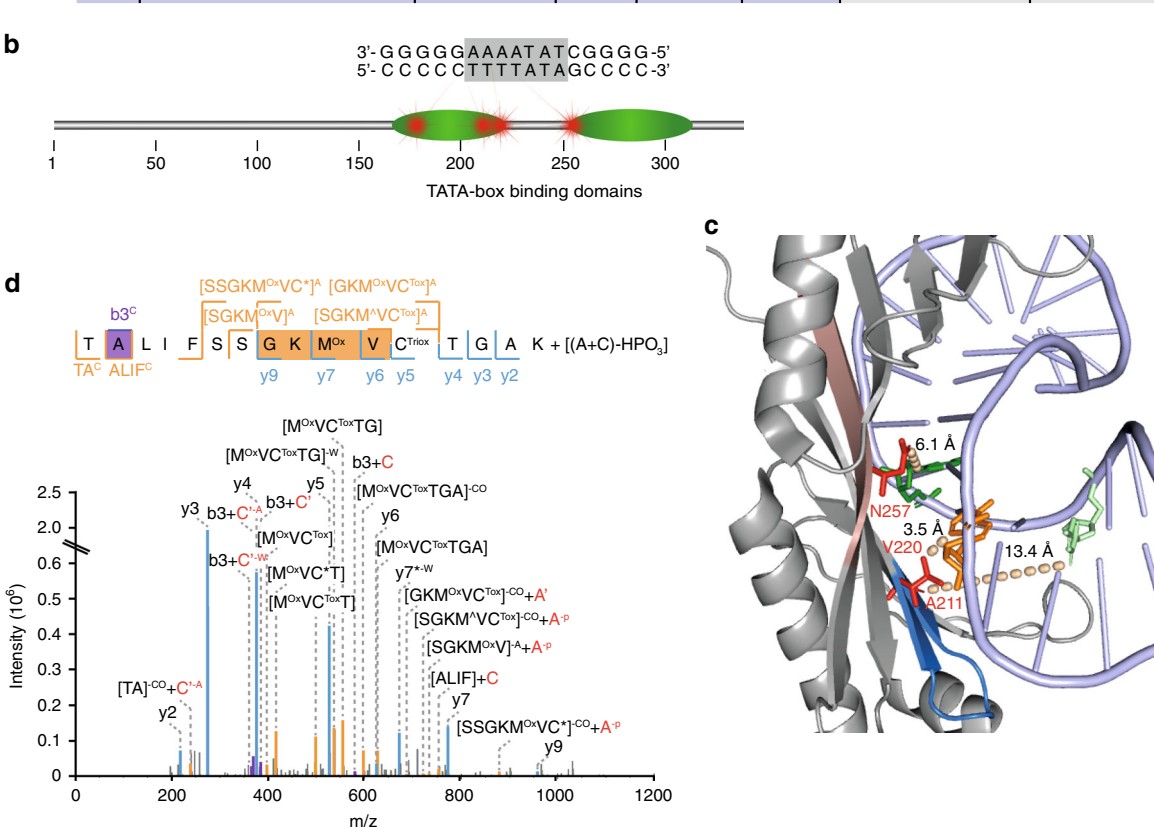

**a**

| DCP id | Peptide sequence | Crosslinked nucleotide | Charge | m/z | Δm [ppm] | Crosslink location | Distance to nucleotide [Å] |
|---|---|---|---|---|---|---|---|
| | fLiX-MS | | | | | Structure Integration | |
| 18 | [209]TTA**LIFSSGKMV**CTGAK[225] | (A+C)-HPO$_3$ | 3 | 773.6829 | 8.5 | A211(C) & V220(A) | 3.5/13.4 |
| 19 | [255]IQ**NM**VGSCDVK[265] | TT-HPO$_3$ | 2 | 902.3474 | 3.4 | N257(T) | 6.1 |
| 20 | [178]**L**DLKTIALR[186] | C-HPO$_3$ | 2 | 635.3821 | 3.4 | L178(C) | 17.2 |

**b**

3'- G G G G G A A A A T A T C G G G G -5'
5'- C C C C C T T T T A T A G C C C C -3'

TATA-box binding domains

**c**

**d**

**Fig. 5 Cross-linking of the TATA-box binding transcription factor TBP. a** Overview of the four identified cross-links on three unique peptides. Blue background indicates information obtained from cross-linking experiments and gray background information obtained from the crystal structure (PDB ID: 1C9B[41]). Red letters indicate possible cross-linked amino acids or cross-linked nucleotides, respectively. **b** Schematic view of the domain structure of TBP, the TATA box (gray shading), and surrounding nucleotides, as well as the cross-links. **c** Crystal structure of TBP bound to an extended Adml promoter (PDB ID: 1C9B[41]). Location of the cross-link of N257 to one of the two thymidines (green) and of the cross-links of amino acid V220 to deoxyadenosine (orange) and A211 to deoxycytidine (light green). Cross-linked amino acids are depicted in red, the peptide [255]IQNMVGSCDVK[265] in light red, and the peptide [209]TTALIFSSGKMVCTGAK[225] in blue. Dashed lines represent the distance of amino acid to nucleotide and distance is shown above the line. **d** MS2 ion series and spectra of the cross-linked TTALIFSSGKMVCTGAK peptide. Abbreviations in the MS2 spectra and MS2 ion series: caret: neutral loss of CH$_4$SO, Tox: trioxidated cysteine (cysteic acid). Other abbreviations as in Fig. 4.

could be a higher flexibility of the TBP–DNA complex in solution, compared with the "frozen" picture of the crystal structure.

An interesting observation in the MS2 spectrum of the [178]LDLKTIALR[186] peptide is that its fragment ions y6, y7, y8, and y9 are exclusively observed with a mass shift of +27.995 Da, corresponding to the addition of carbon monoxide (CO) (Supplementary Fig. 3a). Searching for the source of this adduct, we analyzed all peaks in the lower m/z range and identified a prominent peak at m/z = 89.06 that equaled deoxyribose after loss of CO. Together with a strong marker ion of [deoxycytidine −CO], this provides evidence that the CO adduct is derived from the deoxyribose part of the deoxycytidine, which is additionally cross-linked to the central lysine of the peptide and cut off during HCD fragmentation (Supplementary Fig. 3c). Therefore, we hypothesize that both L178 and K181 were cross-linked to deoxycytidine at the same time and to different parts of the nucleotide.

**Ex vivo fliX-MS in mouse embryonic stem cells (ESCs).** Having established that fliX-MS is highly specific for cross-linking protein–DNA interactions in in vitro assembled protein–DNA complexes, we next asked whether the method could be also applied to cells. To investigate this, we resuspended mouse ESCs (mESCs) in phosphate-buffered saline (PBS) and subjected them to femtosecond UV-laser radiation. We isolated chromatin from the cross-linked cells, following a DNA biotinylation protocol[43], and enriched peptide–DNA cross-links as in the standard fliX-MS workflow (Fig. 6a). Comparison with a nonirradiated control allowed the identification of specific peptide–DNA cross-links.

Analyzing the data with the RNP(xl) software identified several high-confidence cross-links on TFs. Among those, we manually annotated and validated six bona fide cross-links (Fig. 6b, d, e, Supplementary Fig. 4c–e). All cross-links were exclusively present on the DBDs, which once more highlights the specificity of fliX-MS. In addition, fliX-MS was capable to cover different types of protein–DNA interactions, as cross-linked DBDs represented

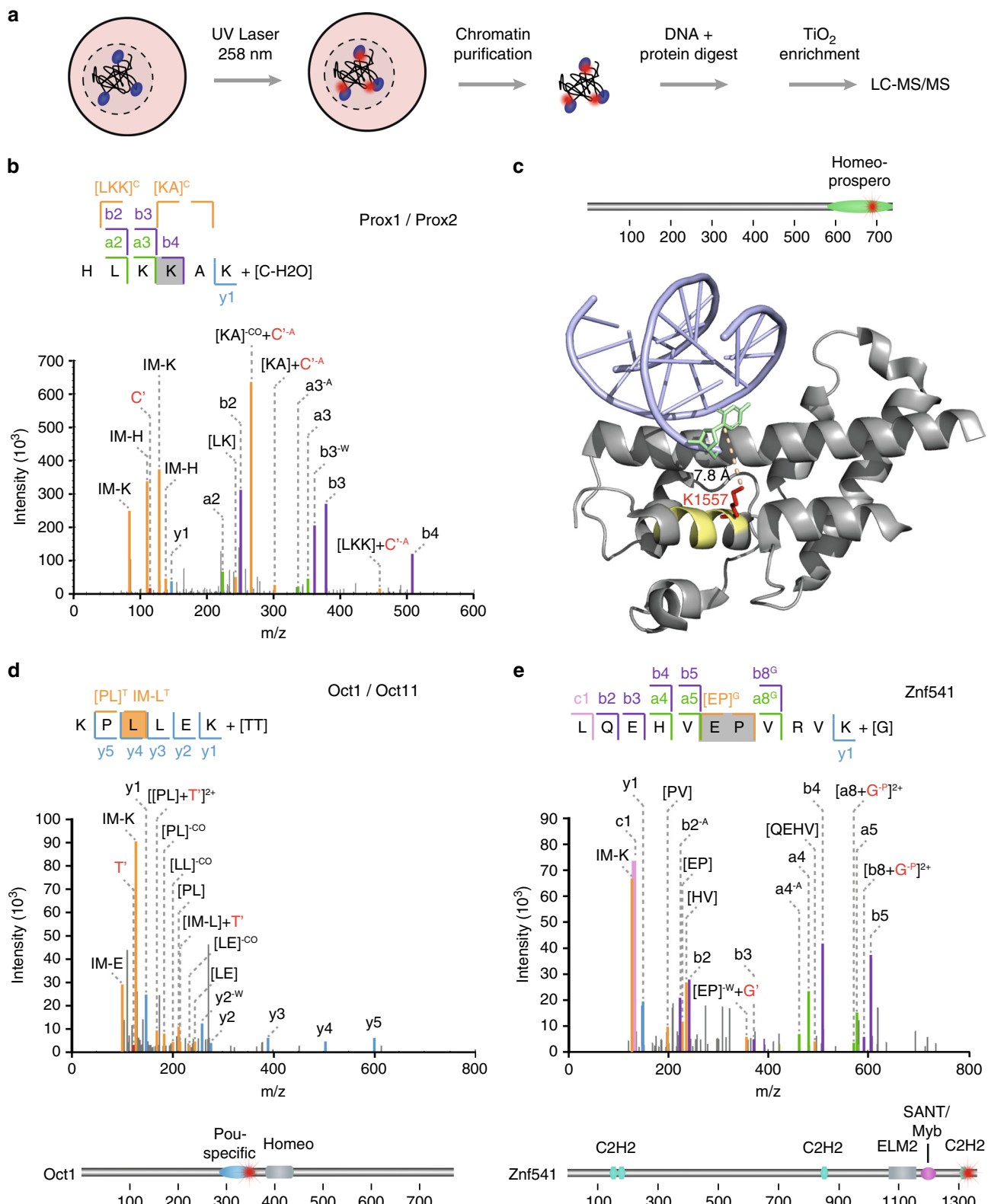

**Fig. 6 fliX-MS applied to mouse embryonic stem cells. a** Schematic overview of the chromatin purification and enrichment of peptide–DNA adducts from laser UV-cross-linked embryonic stem cells. **b** MS2 ion series and spectra of the transcription factor Prox1/2 peptide ⁵⁸⁴HLKKAK⁵⁸⁹ cross-linked to a deoxycytidine monophosphate. Abbreviations as in Fig. 4. **c** Crystal structure of the *D. melanogaster* prospero domain (PDB ID: 1XPX[64]) and location of the ¹⁵⁵²HLRKAK¹⁵⁵⁷ peptide, which is homologous to the cross-linked ⁵⁸⁴HLKKAK⁵⁸⁹ peptide. The peptide ¹⁵⁵²HLRKAK¹⁵⁵⁷ is highlighted in yellow, deoxycytidine in green, and K1557 in red. The distance of K1557 to the cytosine is indicated. **d, e** MS2 ion series of two other high-confidence cross-links of the transcription factor Oct1/Oct11 and the zinc finger protein Znf541. Abbreviations as in Fig. 4. Schematic representation of the cross-link location and domain structure is shown below the respective MS2 spectrum.

four different classes, including homeo–prospero, bHLH, ZNF, and SANT/Myb domains.

The Prox1/2 peptide [584]HLKKAK[589] was cross-linked to a deoxycytidine monophosphate via K587 and is part of the DNA-binding homeo–prospero domain (Fig. 6b, c). Since this domain is fairly large, we wondered whether the interaction would agree with known structural data. Locating the [584]HLKKAK[589] cross-link in the crystal structure of the highly conserved prospero protein in *D. melanogaster* (Supplementary Fig. 4a), we observed that the *Drosophila* counterpart peptide ([1552]HLRKAK[1557]) is in an alpha helix in close vicinity to the DNA, where K1557 points toward the deoxycytidine with a distance of 7.8 Å (Fig. 6c). This demonstrates that the ex vivo generated cross-link specifically reflected a TF–DNA binding event.

The peptide KPLLEK was cross-linked to a dithymidine and could be mapped to several different TFs, namely Oct1, Oct2, Oct11, and Hes2, as well as to the mitotic spindle assembly checkpoint protein Mad2l2 (Fig. 6d). Analyzing the proteome of the same murine ES cell line to a depth of >9700 proteins (Supplementary Fig. 4b) revealed exclusive expression of Oct1 and Oct11 in this dataset, suggesting that the cross-linked peptide is derived from one of the two proteins. In both cases the peptide forms part of the conserved Pou-specific DBD, again underlining the feasibility of fliX-MS to identify functional ex vivo protein–DNA contacts.

The high-confidence cross-linked peptides of Znf541, Smarca1, Zfp91, and Znf354c supported this further (Fig. 6e, Supplementary Fig. 4c–e). As for the other two ex vivo cross-links, our data defined both the exact cross-linked nucleotide, as well as the amino acid position with a precision of maximum two adjacent amino acids. Of technical note, the spectrum of Znf541 contained a rare C1 ion, which can be formed during HCD fragmentation of peptides with an asparagine or glutamine in second position[44].

## Discussion

Although interaction of TFs with DNA is a hallmark of gene transcription, it has remained an understudied area of biology due to several technical limitations: (i) Current methodologies such as chromatin immunoprecipitation followed by next-generation sequencing (ChIP-Seq) or proteomics (ChIP-MS) cannot differentiate between direct DNA binding and co-recruitment via other DNA-binding proteins. (ii) Direct TF–DNA binding assays depend on the availability of recombinant proteins and do not necessarily reflect DNA binding in living cells. (iii) Cocrystallization or NMR of protein–DNA complexes are highly laborious and not even possible for many TFs. Hence, a tool to directly assign protein–DNA interactions with amino acid and nucleotide resolution would have a strong impact on biological research.

High-intensity femtosecond lasers provide a plethora of applications reaching from ultrafine material processing[45], high-precision medical surgery[46], to the detection of biomolecular processes[47]. In the search for effective cross-linking methods of proteins and DNA, we and others have previously shown that femtosecond lasers are promising for this purpose because they provide high cross-linking yields while minimizing DNA damage[20,24,48,49]. With recent advances in XL-MS in the sample preparation, MS instrumentation, and bioinformatics side[17,50], we here combined this highly effective cross-linking strategy with an optimized purification protocol for cross-linked peptides, and MS-based read out of protein–DNA cross-links. Our method can map protein–DNA interactions both in vitro as well as in cells, making it a powerful tool for many different research topics.

As a proof of principle, we applied our femtosecond laser-induced cross-linking followed by high-resolution MS (fliX-MS)

pipeline to in vitro assembled nucleosomes, as well as to recombinant TFs. Notably, we were able to detect cross-links to all four DNA bases. For recombinant TFs, all cross-links mapped exclusively to annotated DBDs, providing confidence for future applications of fliX-MS for the de novo identification of protein–DNA interactions. Although UV cross-linking in addition to DNA–protein cross-links also produced protein–protein cross-links, the observed DNA–protein cross-links strongly depended on a specific DNA-consensus site, suggesting that femtosecond UV-laser irradiation does not interfere with the protein conformation.

One technical limitation of the current fliX-MS workflow is the dependency on enzymatic protein digestion for MS analysis. In case of the nucleosome, many of the annotated amino acid–DNA contacts locate in regions, which are enriched in lysine and arginine residues and the resulting peptides are often too short to be measurable by LC–MS/MS. For instance, histone H2A has seven annotated DNA-binding sites (R30, R33, R36, K37, R43, K75, and R78, Interpro: P04908) in regions where tryptic digestion would produce peptides less than seven amino acids in length, which are difficult to observe by MS analysis. This limitation could be overcome by the use of enzymes with different specificity such as Arg-C or chymotrypsin, or by chemically modifying all lysine residues in the protein complex, which is commonly applied for the analysis of histone posttranslational modifications by MS[51].

Apart from localizing protein regions, our method revealed detailed structural information of DNA–protein interactions, especially where no crystal structure was available. Despite being one of the first studied DNA-binding proteins[52], mechanistic information on DNA interaction of nuclear factor 1/C (NF1) has been limited to mutation[33] and truncation[26] studies, as well as DNA-binding analyses in combination with modified bases[31]. Notably, our fliX-MS data on the NF1–DNA complex were in close agreement with the previous biochemical data. All cross-links were in the subregion of the CTF/NF1 binding domain that was reported to confer sequence-specific DNA-binding activity[26], while no cross-link was found in the remaining part of the CTF/NF1 binding domain that mediates only unspecific DNA binding. Furthermore, two cross-linked amino acids were in close vicinity to mutation sites that had been shown to reduce or eliminate DNA binding[33]. Taking advantage of the sequence information provided by the cross-linked di- and trinucleotides, we explicitly localized the cross-links on the NF1 consensus sequence in four out of five cases, confirming the interaction with both DNA strands originally proposed of early NF1–DNA contact site analyses[31]. In addition, we revealed interactions of NF1 with the cytosines on the TTGGA reverse strand, which have not been observed before. Given the detailed information of binding contacts from our experiments, molecular modeling of the NF1–DNA complex might now be feasible. In fact, the CTF/NF1 domain shares structural homology with the structurally resolved SMAD DBD[53]. With the additional information gained by fliX-MS, we envision that the structure of the NF1–DBD in complex with DNA can be finally resolved.

Comparing our data on recombinant nucleosomes and TBP bound to its target DNA with the respective crystal structures showed that the peptide–DNA cross-links were largely in agreement with the intramolecular distances in the electron density maps. However, three cross-links of the nucleosome, and two of TBP revealed distances between amino acid side chains and nucleotides that were too large (>16 Å) to support a direct contact according to the crystal structure. The most likely explanation is that our method is capable to detect different conformational states of protein–DNA complexes in solution, while crystal structures reflect only a single discrete structural conformation. In

support, cryo-EM studies on nucleosomes[29,30] revealed a large degree of structural dynamics, based on partial unwrapping of the DNA, also known as DNA breathing. Notably, all distant cross-links lie on an H2A helix, which was described to be especially susceptible to conformational rearrangements in the nucleosome[29,30]. In case of TBP, the two distant cross-links all pointed to the same nucleotide, namely the first cytosine downstream of the TATA box on the reverse strand. TBP binding to the DNA requires significant DNA deformation, including opening of the minor groove and a reduction of the helical twist[9,54]. To generate the cross-links identified here, the DNA must be able to take up a much stronger deformed conformation than the crystal structure would suggest. Taken together, this demonstrates that fliX-MS is capable to add additional information to crystal structure data, by providing evidence for structural flexibility of certain subregions.

Having established the potential of fliX-MS to accurately map DNA-binding contacts in vitro, we were encouraged to also extend our cross-linking strategy to cells. Despite a potential for optimizing both chromatin enrichment efficiency and MS sensitivity much further, we were able to identify several bona fide examples of TF–DNA cross-links. Reassuringly, these cross-links were all located on DBDs, suggesting that fliX-MS can indeed identify specific protein–DNA interactions in cells. In addition, our method might be also applicable to DNA-pulldown experiments[55], after laser irradiation of the eluted protein–DNA complexes. This would be especially useful for analysis of selected TFs, which cannot be expressed recombinantly.

In conclusion, we have developed a workflow to map protein–DNA contacts in both in vitro and cellular contexts. Given the scientific importance of such contacts, we believe that fliX-MS will have major impacts in many fields of biology and even clinical research. Current developments on both MS technology and data analysis side may even allow the mapping of global DNA interactomes in near future.

## Methods

**Protein expression and purification**. For the reconstitution of recombinant human nucleosomes, histone proteins (H3.1, H4, H2A, H2B) were expressed in *E. Coli* BL21 (DE3) cells and purified via inclusion body preparation, denaturing gel filtration, and ion exchange chromatography[56–58] (see Supplementary Fig. 5). The pUC19-16×601 plasmid was amplified in *E. Coli*, the 601 strong positioning DNA sequence excised by digestion with EcoRV and purified by PEG precipitation[58]. Purified DNA was further digested with EcoRI and biotinylated with biotin-11-dUTP (Jena Biosciences) using Klenow fragment (3′ → 5′ exo-) polymerase (NEB). Finally, histones were refolded into octamers and nucleosomes reconstituted by salt gradient dialysis[56].

6xHis-tagged recombinant NF1 was cloned into a baculovirus vector, expressed in Sf9 cells, and purified by nickel column chromatography[59].

Recombinant TBP was purchased from Active Motif (81114).

**Assembly of protein–DNA complexes**. Sequences of the DNA oligonucleotides used for in vitro experiments were: NF1: 5′-AAT TCC TTT TTT TGG ATT GAA GCC AAT CGG ATA ATG AGG-3′ (sense, wild type), AAT TCC TTT TTT TGC GCT AAA GCG TAG TGG ATA ATG AGG (sense, mutant) for all experiments except Fig. 2d, 5′-AAG TCC TTT TTT AGG ATT GAA GCC AAT CGG CTG ATG AGG-3′ (sense, wild type), 5′-AAG TCC TTT TTT AGC GCT AAA GCG TAG TGG CTG ATG AGG (sense, mutant) for Fig. 2d; TBP: 5′-CCT GAA GGG GGG CTA TAA AAG GGG GTG GGG GCG CG-3′ (sense, wild type), 5′-CCT GAA GGG GGG CTG TAA AAG GGG GTG GGG GCG CG-3′ (sense, mutant). For each sequence both sense and antisense oligonucleotides were synthesized with a biotin covalently linked to the 5′-end. Double-stranded DNA probes were generated by incubating 100 pmol of each sense and antisense oligo in 25 µl of annealing buffer (10 mM Tris-Cl pH 7.5, 50 mM NaCl, 1 mM EDTA) at 95 °C for 5 min followed by cooling down to room temperature for 60 min. For all western blots and fliX-MS experiments other than Fig. 2d, 13 µg NF1 protein (234 pmol) was incubated with 30 pmol of annealed DNA for 25 min at room temperature in 50 µl NF1 binding buffer (90 mM NaCl, 0.5 mM EDTA, 5% glycerol, 0.55 mM β-mercaptoethanol, 5 mM Tris-Cl (pH 8.0), 1 µg BSA). For the western blot in Fig. 2d, 4.2 µg NF1 (74 pmol) was incubated with 14 pmol of annealed DNA for 25 min at room temperature in 50 µl NF1 binding buffer containing 200 mM NaCl. For all experiments with TBP 15 µg (380 pmol) of recombinant protein was

incubated with 30 pmol of DNA for 25 min at RT in 200 µl TBP binding buffer (NF1 binding buffer + 2 mM MgCl₂). For electrophoretic mobility shift assays (EMSA), 0, 25, 50, 90, or 125 ng NF1 (corresponding to 0, 442, 885, 1592, and 2212 fmol) were incubated for 25 min with 200 fmol DNA in 20 µl NF1 binding buffer containing 200 mM NaCl for 25 min at room temperature.

**Electrophoretic mobility shift assay**. 5× TBE Hi-Density buffer (15% Ficoll (w/v), 5% glycerol (v/v), 1× TBE Buffer (Invitrogen, 15581044)) was added in a 1:4 ratio to assembled protein–DNA complexes and samples separated on a 6% DNA retardation gel (Invitrogen) at 100 V in 0.5× TBE buffer for 45 min. The gel was incubated with 3 µl SYBR Green I (Sigma-Aldrich) diluted in 30 ml of 1× TBE buffer and rocked for 20 min at room temperature. Excess SYBR Green I dye was washed off by rinsing the gel three times with MilliQ water. DNA was visualized on a LAS4000 Image Quant (GE Healthcare).

**Western blot**. For DNase I and proteinase K experiments from Fig. 2e, f, samples were diluted fourfold to final concentrations of 10 mM Tris-Cl (pH 8.0), 2.5 mM MgCl₂, and 0.5 mM CaCl₂. One microliter of DNase I or one microliter of proteinase K was added for digestion experiments and left at 37 °C (DNase I and untreated) or 56 °C (proteinase K) for 1.5 h before denaturation. All other cross-linked samples were denatured directly before denaturing gel electrophoresis. To this end, 4× LDS sample buffer (Invitrogen, NP0007) was added to the cross-linked samples in a 1:3 ratio and samples boiled for 5 min at 95 °C. Proteins were separated on a NuPAGE 4–12% Bis-Tris Gel (Invitrogen) at 150 V for 45 min in 1× MOPS Running Buffer (Invitrogen) and transferred onto a nitrocellulose membrane (GE Healthcare) at 75 V for 90 min at 4 °C in 1× blotting buffer (25 mM Tris-Cl, 192 mM glycine (pH 8.3), 20% methanol). For detection of biotin-labeled DNA, blots were blocked with 15 ml blocking buffer (Active Motif EMSA kit, 37341) for 15 min and incubated with 50 µl streptavidin–HRP conjugate (Active Motif EMSA kit, 37341) in 15 ml blocking buffer for 15 min. Blots were washed three times with 10 ml TBS-T buffer (150 mM NaCl, 50 mM Tris-Cl (pH 7.6), 0.1% Tween-20). Fifteen milliliters Substrate Equilibration Buffer (Active Motif EMSA kit, 37341) was added and blots incubated at RT for 5 min. DNA was visualized by incubation with chemiluminescent reagent (WESTAR ηC 2.0, Cyanagen) for 1 min and imaged on the LAS4000 Image Quant. For the detection of 6xHis-tagged NF1 protein, blots were blocked with western blocking buffer (5% skim milk powder in 1× TBS-T buffer) for 45 min followed by incubation with anti-6xHis antibody (MA1-21315, Invitrogen) diluted 1:2000 in western blocking buffer overnight at 4 °C. Blots were washed three times for 10 min with 1× TBS-T buffer and incubated with HRP-coupled anti-mouse IgG (GE Healthcare, NA931) antibody, diluted 1:4000 in 0.5× western blocking buffer (2.5% skim milk powder in 1× TBS-T buffer), for 1 h at room temperature. Blots were washed three times for 5 min with 1× TBS-T, incubated with chemiluminescent reagent for 1 min, and visualized on the LAS4000 Image Quant. For membrane stripping, blots were washed twice in TBS and incubated 10 min with 10 ml of Restore PLUS Western Blot Stripping Buffer (Thermo) at room temperature and washed four times with TBS. Band intensities were quantified using ImageJ version 1.52a. All blots are shown in Supplementary Figs. 1 and 6.

**Femtosecond laser-induced cross-linking**. For UV irradiation, a femtosecond fiber laser (active fiber systems GmbH, Jena, Germany) with a wavelength of 1030 nm, doubled to 515 nm, was used with a pulse duration of about 500 fs. The wavelength was further doubled to 258 nm using a BBO crystal (Laser components GmbH, Olching, Germany). The laser average power is limited for repetition rates of 0.5–20 MHz, which was adjusted to 0.5 MHz to provide sufficiently high pulse energy for frequency conversion. The laser beam diameter was adjusted to 2.5 mm (e⁻²), in order to fit the inner diameter of a 1.5 ml Eppendorf tube. For a typical pulse energy of 50 nJ (or 25 mW average power), this diameter results in a peak intensity of 4 MW cm⁻² on the beam axis.

For fliX-MS experiments, 100 µl (45.9 µg) of recombinant human nucleosomes, 100 µl assembled NF1–DNA complex, or 200 µl assembled TBP–DNA complex (see above) were irradiated with 1.25 J total energy and a pulse energy of 50 nJ (25 Mio. pulses with 500 fs pulse length), or left untreated as control.

For UV radiation titration experiments, 25 µl of NF1–DNA complex was irradiated with varying settings as mentioned in the text. For ex vivo cross-linking experiments, 20 Mio. mESCs (E14TG2a) were resuspended in 100 µl PBS and irradiated with 2.1 J total energy and 42 nJ pulse energy.

**Digestion of in vitro samples**. Individual enrichments were performed with 75 pmol of cross-linked NF1–DNA complex (molar amount of DNA), 150 pmol of cross-linked TBP–DNA complex, or 200 µg of cross-linked recombinant mono-nucleosomes (containing 91.8 µg of histone octamer and 941 pmol of DNA), each pooled from multiple UV radiation samples. Urea and Tris-Cl (pH 8.0) were added to the cross-linked samples to final concentrations of 4 M and 50 mM, respectively. After 5 min incubation, urea was diluted to 1 M with 50 mM Tris-Cl (pH 8.0). CaCl₂ was added to 5 mM and MgCl₂ to a final concentration of 2 mM. One microliter of MNase (New England Biolabs, M0247S), one microliter of DNase I (New England Biolabs, M0303S), and three microliters of Benzonase (Merck Millipore, 70746) were added to every 150 pmol of DNA. DNA digestion was

carried out for 90 min at 37 °C. Trypsin and Lys-C were added at a ratio of 1:40 (w/w) compared with the protein amount and incubated for 30 min at 37 °C, followed by overnight incubation at 25 °C. The next day, formic acid (FA) was added to 0.1% final concentration.

**Purification of chromatin associated proteins**. Chromatin extraction and purification from cross-linked mESCs was performed by adapting a published chromatin biotinylation protocol[43]. Three cross-linked or three non-cross-linked control cell pellets, respectively, were resuspended in 300 µl cell lysis buffer (20 mM Tris-Cl (pH 8.0), 85 mM KCl, 0.5% NP-40, 1× PIC (Roche cOmplete)) and immediately centrifuged for 5 min at 2000 × g and 4 °C. Pellets were resuspended in 300 µl SPC-NEB buffer (1 M sorbitol, 50 mM Tris-Cl (pH 8.0), 5 mM CaCl$_2$, 1× PIC) and incubated at 37 °C for 1 min. Six microliter of MNase (New England Biolabs) was added and samples incubated for 13 min at 37 °C. EDTA was added to a final concentration of 50 mM. Nuclei were pelleted by centrifugation at 2000 × g and 4 °C for 5 min. After resuspension with 300 µl 0.2% SDS buffer (0.2% SDS, 20 mM Tris-Cl (pH 8.0), 1 mM EDTA pH 8.0, 1× PIC), samples were sonicated in a Bioruptor (Diagenode) at a low setting for three cycles, 30 s on/30 s off. After centrifugation at 8600 × g and 4 °C for 5 min, the supernatant was removed and dialyzed twice (once overnight and once for 6 h) using a 10,000 MWCO membrane (Slide-A-Lyzer, Thermo Fisher) against 3 l of NEB buffer 2 (50 mM NaCl, 10 mM Tris-Cl (pH 7.9), 10 mM MgCl$_2$, 1 mM DTT). BSA was added to 100 µg ml$^{-1}$ and chromatin diluted with NEB buffer 2 (+100 µg ml$^{-1}$ BSA) to a concentration of about 1 µg µl$^{-1}$. Forty-five microliters of T4 PNK (New England Biolabs, M0201S) and five microliters NEBuffer 2 (+100 µg ml$^{-1}$ BSA) were added to 45 µg of cross-linked or non-cross-linked chromatin, respectively. Samples were incubated for 15 min at 37 °C. For the biotin-replacement synthesis, the following reagents were added to 45 µg of cross-linked or 45 µg non-cross-linked chromatin at a concentration of 0.5 µg µl$^{-1}$, respectively: 3.1 µl of 10 mg ml$^{-1}$ BSA (New England Biolabs), 21 µl of 10× NEB buffer 2 (New England Biolabs), 76.3 µl of 0.4 mM Biotin-dATP (Jena Biosciences), 76.3 µl of 0.4 mM Biotin-dCTP (Jena Biosciences), 3.1 µl of 10 mM dTTP/dGTP (New England Biolabs), and 30 µl of T4 Polymerase at a concentration of 3 U µl$^{-1}$ (New England Biolabs, M0203S). After incubation at 12 °C for 15 min, EDTA was added to a final concentration of 50 mM. Next, the chromatin was dialyzed overnight against 3 l of dialysis buffer (50 mM Tris-Cl (pH 7.5), 1 mM EDTA, 150 mM NaCl, 1× PIC) at 4 °C. GdmCl denaturation buffer (8 M guanidine hydrochloride (GdmCl), 13.33 mM TCEP, 133.33 mM Tris-Cl (pH 8)) was added in a ratio of 3:1 and samples were boiled for 10 min at 99 °C. After allowing samples to cool down to room temperature, chloroacetamide was added to 40 mM and incubated for 20 min. 1.4 mg of T1 streptavidin beads (Thermo Scientific) were equilibrated by washing once with 1 ml of 1× B&W buffer (5 mM Tris-Cl (pH 7.5), 0.5 mM EDTA, 1 M NaCl) and once with 1 ml of 1× GdmCl wash buffer (0.6 M GdmCl, 1 mM TCEP, 10 mM Tris-Cl (pH 8)). Chromatin samples were diluted tenfold with 25 mM Tris-Cl (pH 8) and added to the beads. After incubation for 90 min at room temperature, beads were washed by incubating the beads 15 min with 1 ml of GdmCl wash buffer (0.6 M GdmCl, 10 mM Tris-Cl (pH 8)) for three times. After two washes with 1 ml of BW2× buffer (10 mM Tris-Cl (pH 8.0), 1 mM EDTA, 0.1% Tritone-X100, 2 M NaCl), two washes with 1 ml of SDS wash buffer (25 mM Tris-Cl (pH 8.0), 1 mM EDTA, 1% SDS, 200 mM NaCl), and two washes with TBS buffer (150 mM NaCl, 50 mM Tris-Cl (pH 7.6)), beads were resuspended in 50 µl MNase/Benzonase digestion buffer (2 mM MgCl$_2$, 5 mM CaCl$_2$). DNA was digested by the addition of 1 µl of MNase (New England Biolabs), 1 µl of DNase I (New England Biolabs) and 3 µl of Benzonase (Merck Millipore) and incubation for 90 min at 37 °C. GdmCl was added to 0.6 M and Tris-Cl (pH 8.0) to 10 mM. Two microliters of trypsin (0.5 µg µl$^{-1}$) and Lys-C (0.5 µg µl$^{-1}$) were added and proteins digested overnight.

**Enrichment of DNA-cross-linked peptides**. Peptides were desalted on StageTips containing C18 material (3× C18 disks) (Empore)[60]. StageTips were equilibrated sequentially with 100 µl methanol, 100 µl buffer B3 (95% acetonitrile (ACN)/0.1% FA), 100 µl buffer B2 (80% ACN/0.1% FA), 100 µl buffer B1 (50% ACN/0.1% FA), and 100 µl buffer A (0.1% FA). Samples were loaded and washed twice with buffer A. Peptides were eluted twice with 50 µl buffer B1 and once with 50 µl buffer B2. Eluates were combined and dried on a centrifugal evaporator. Three hundred microliters of TiO$_2$ blocking buffer (60% ACN, 0.1% trifluoroacetic acid (TFA), 300 mM lactic acid) was added and samples resuspended at 25 °C and 2000 rpm for 5 min. Fifteen milligrams of TiO$_2$ beads (GL Sciences) were resuspended in 25 µl of buffer B2 and added to the sample. After 5 min incubation at 25 °C and 2000 rpm, beads were pelleted by centrifugation at 2000 × g for 1 min. Beads were washed once with TiO$_2$ blocking buffer (centrifugation at 2000 × g, 1 min) and three times with buffer B2 (centrifugation at 2000 × g, 1 min). Beads were resuspended with 100 µl of buffer B2 and loaded onto C8 StageTips (3× C8 disks, Empore). Beads remaining in the tube after the first transfer were resuspended once more with 100 µl of buffer B2 and loaded onto the same C8 StageTip. Peptide–nucleotide cross-links were eluted twice with 40 µl TiO$_2$ elution buffer (60% ACN/5% NH$_4$OH) and samples dried on a centrifugal evaporator. Samples were dissolved in 5 µl buffer A* (2% ACN/0.1% TFA) for MS analysis.

**LC–MS/MS analysis**. Online chromatography was performed with a Thermo EASY-nLC 1200 UHPLC system (Thermo Fisher Scientific, Bremen, Germany) coupled online to a Q Exactive HF-X mass spectrometer with a nano-electrospray ion source (Thermo Fisher Scientific). Analytical columns (50 cm long, 75 µm inner diameter) were packed in-house with ReproSil-Pur C18 AQ 1.9 µm reversed phase resin (Dr. Maisch GmbH, Ammerbuch, Germany) in buffer A (0.1% FA). During online analysis the analytical columns were placed in a column heater (Sonation GmbH, Biberach, Germany) regulated to a temperature of 60 °C. Peptide mixtures were loaded onto the analytical column in buffer A and separated with a linear gradient of 5–20% buffer B (80% ACN and 0.1% FA) for 50 min, and 20–30% buffer B for 10 min, at a flow rate of 250 nl min$^{-1}$. MS data were acquired with a Q Exactive HF-X instrument programmed with a data-dependent top 12 method in positive mode using Tune 2.9 and Xcalibur 4.1. The S-lens RF level was 40.0 and capillary temperature was 250 °C. Full scans were acquired at 60,000 resolution with a maximum ion injection time of 20 ms and an AGC target value of 3E6. Selected precursor ions were isolated in a window of 2.0 *m/z*, fragmented by HCD with normalized collision energies of 30 for in vitro complexes and 35 for samples derived from cell cross-linking), and measured at 15,000 resolution with maximum injection time of 60 ms and AGC target of 1E5 ions. Precursor ions with unassigned or single states were excluded from fragmentation selection and repeated sequencing minimized by a dynamic exclusion window of 20 s.

**Data analysis**. Raw MS data were analyzed using the RNP(xl) workflow from the OpenMS Nodes v2.0.3 package implemented in the Proteome Discoverer software (v. 2.1.1.21)[17,50]. Control and UV-irradiated files were aligned by the retention time and precursors present in both conditions removed[17]. For in vitro fliX-MS experiments, searches were performed with modified Uniprot databases for human (nucleosomes and TBP) and pig (NF1), in which isoforms of the recombinant proteins were removed. Ex vivo fliX-MS data were searched against the Uniprot database for mouse (mESCs) in combination with contaminant sequences from the MaxQuant software package[61]. FDR control was performed by searching against a target-decoy version of the respective database. For in vitro flix-MS data, oxidation of methionine, trioxidation of cysteine (cysteic acid), and carbamidomethylation of lysine and N-termini were allowed as variable modifications. For ex vivo flix-MS data, oxidation of methionine was defined as variable modification and carbamidomethylation of cysteines as static modification. The maximum allowed number of missed cleavages was 2 in all cases. Precursor DNA modifications were searched against all possible combinations of up to four connected nucleotides and possible modifications of −H$_2$O, −HPO$_3$, −H$_3$PO$_4$, and +HPO$_3$. Precursor mass tolerance was set to 10 ppm and fragment mass tolerance to 20 ppm. Incremental masses of shifted ions were set in the following order: nucleotide, nucleotide −H$_3$PO$_4$, −HPO$_3$, −H$_2$O, nucleobase, nucleobase −NH$_3$, and nucleobase −CO (only thymine).

Manual curation of spectra proposed by RNP(xl) was performed as follows: (i) Precursor ions were evaluated for the correct assignment of the charge state and monoisotopic peak. (ii) The corresponding MS2 spectra were evaluated for >40% amino acid coverage combining a, b, and y ions. (iii) If the mass shift on the precursor ion reflected more than one nucleotide, nucleotides were required to be observed as marker ions in the low mass range. (iv) High-intensity fragment ions, which did not represent the unmodified peptide sequence, needed to be explainable by the DNA cross-link. RNP(xl) automatically annotates a, b, and y ions and immonium ions shifted by a nucleotide or nucleobase. In addition, all spectra were further analyzed for shifted and nonshifted internal ions using ProteinProspector (v 5.24.0). Supplementary Data 2 lists the identified a, b, and y ions and mass-shifted ions with additional information for all spectra.

Analysis of crystal structures and validation of the cross-links in crystal structures was performed using PyMol (the PyMOL Molecular Graphics System, version 1.2r3pre, Schrödinger, LLC).

**Reporting summary**. Further information on research design is available in the Nature Research Reporting Summary linked to this article.

## Data availability
The MS proteomics data have been deposited to the ProteomeXchange Consortium[62] (http://proteomecentral.proteomexchange.org) via the PRIDE partner repository[63] with the dataset identifier PXD014898. All other data are available from the corresponding authors on reasonable request.

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

## Acknowledgements

We thank Henning Urlaub and Alexandra Stützer from the Max Planck Institute for Biophysical Chemistry for advice on data analysis. We thank Till Bartke and Benjamin Foster from the Institute of Functional Epigenetics (Helmholtz Zentrum München) for providing expression plasmids and assistance in generating recombinant nucleosomes. We thank our colleagues at the Department of Proteomics and Signal Transduction for help and fruitful discussions. We thank Alexander Strasser, Igor Paron, and Christian Deiml for excellent technical assistance.

## Author contributions

M.W., C.R., and A.R. designed the study; A.R. performed all experiments and data analysis; R.A. and R.K. conducted laser irradiation; J.F.-M. prepared recombinant NF1 protein; A.R. and M.W. wrote the manuscript; M.W., C.R., M.M., S.N., and M.B. supervised the study.

## Competing interests

The authors declare no competing interests.
