## [Peer Review File · Nature Communications]

Reviewers' comments:

Reviewer #1 (Remarks to the Author):

Overview:

More than thirty years ago pioneering studies by the J.T. Lis laboratory in *E. coli* and *Drosophila* (Gilmour and Lis, 1984, PNAS 81; Gilmour, D.S. et al, 1986, Cell 44) demonstrated the potential power of UV irradiation mediated protein DNA crosslinking for studying the biology of DNA binding proteins. As a matter of fact these papers are also the first ones describing a kind of chromatin IP assay, which is nowadays in most of the cases carried out employing formaldehyde (FA) crosslinking. Follow-up work by the laboratories of M.D. Biggin, D. Houde and G.L. Hager (Walter, J. and Biggins, M.D., 1997, Methods 11; Moss, T. et al., 1997, Methods 11; Nagaich, A.K. et al., 2004, Mol. Cell 14) demonstrated that the efficiency of UV DNA protein crosslinking can be increased by utilizing pulsed UV light emitted from lasers. Concomitantly, literature reports from the last 20 years also suggest that the technological evolution of laser technology (e.g. the move from nanosecond to picosecond laser pulsing technology) bear the potential to address the major shortcoming of UV mediated DNA protein crosslinking, namely its low efficiency that comes along with extremely low recovery of covalently bound DNA protein complexes hampering the general applicability of the approach.

This is in stark contrast to the field of RNA protein interactions for which UV crosslinking has been and still is the mainstay of any genomics or proteomics follow-up methodology. Until now, the huge difference in the yield of UV-crosslinked RNA-protein versus DNA-protein complexes is explained by the compromised photo activation of the nucleobases in the double stranded DNA (base pairing, base stacking) versus the situation found in single stranded RNA. Counteracting this by prolonging the exposure time (using standard UV lamps) is not feasible because under such conditions severe photo damage of biological macromolecules prohibits subsequent biochemical workflows (e.g. cells cannot be efficiently lysed anymore).

Therefore, FA crosslinking, which exhibits recovery of DNA-bound proteins in the range between 0.01-2.0% and that forms metastable zero-length covalent DNA-protein complexes (but is partially also recovering piggy-back/indirect DNA-protein interactions) is currently the commonly employed starting point of any large scale experiments like ChIP-seq, FAIRE-seq and Hi-C/3C-seq.

In this context the proposed manuscript by Reim and colleagues tries to overcome the current bottlenecks of examining DNA-protein interactions with the help of UV irradiation by introducing the application of femtolaser pulse technology in conjunction with state-of-the-art nanoLC-MS analysis of DNA-crosslinked tryptic peptides. At first the authors claim extremely high crosslinking efficiency (up to 100%) in reconstituted *in vitro* systems but the efficiencies of crosslinking in live cells as well as quantifying the ability to recover the UV XLinked molecules are not addressed.

Nevertheless, this is with no doubt a very interesting and pioneering study which should be published in a decent journal like Nature Communications. Emphasizing the challenging task of mapping the interaction surface between proteins and DNA, this is much needed for the field. The used approach is innovative, appropriate and a step in the right direction but for reasons outlined below a couple of issues mainly related to (quality) control experiments, data representation, the need for a much better and detailed description of the methods and a more critical discussion have to be resolved in a major revision prior to publication of the work.

In addition and since this is important for the editor to judge future citation of this study, the general availability and robustness (easy to replicate?) of the experimental femtolaser setup is currently not clear at all. In other words, could Wierer, Reim and coworkers envision that their technology will become available on a routine basis in the near future?

Therefore, I would like to recommend the work of Reim and colleagues for publication in Nature Communications given that the important concerns rose above and detailed below can be rectified in form of a major revision.

Major points:

Description of methods: the materials and methods section is lacking information that is crucial for both understanding the paper and the possibility to reproduce the work.

☐ In case of the synthetic oligonucleotides it is not clear if the 5'-biotinylation is only present at the "sense" strand or on both (also reverse) oligos?

☐ In general, it is difficult to extract information concerning the concentration and total amount of protein(s) and DNA in the reconstituted in vitro systems (NFI, TBP, reconstituted nucleosome core particles)

☐ The LC/MS methods do not provide the following parameter settings: source temperature, RF-value for S lense (or equivalent of it), target value and injection time for MS1

☐ The description of the data analysis is lacking the following parameter description: number of max. allowed missed cleavages, PD and RNPxl version and the criteria applied for manual curation (e.g. what is the percentage of spectra initially proposed by PD/RNPxl that had been manually removed?)

☐ The sketch of the femtolaser setup (see Fig. 1) and its description makes it complicated to compare the work presented by Reim et al. with the literature (picolaser or nanolaser configurations etc.). Please indicate the pulse repetition rate (is it 0.5 MHz?), the origin/manufacturer of the b-barium borate crystal and the configuration of the active fiber systems laser ("high-repetition rate" or "high

pulse energy"). It would be very beneficial for the general reader if the authors could provide in the main text an idea on how much average power (Watt) and peak power (Watt) is actually applied to the samples (also W/cm²). Moreover, it would be nice to clarify (for their standard parameters of 500 fs pulses at 50 nJ and a total energy of 1.25 J) how many pulses (equal to the total exposure time of the biological samples) are actually required (about 5,000? ~10 ms exposure time?). I am asking this because until now it seems that there is no standard in presenting such data and older literature is often using W and W/cm² in the figures. Finally estimation on how many photons per nucleotide are actually needed to reach saturating photo crosslinking of NFI to the synthetic DNA would be quite nice to enhance the understanding of the work by non-physicists.

Data availability: the data are uploaded to the EBI PRIDE data base but unfortunately no reviewer login credentials have been communicated. Please allow the reviewers to access the .RAW files during the revision process. Additionally, it would be helpful to provide the following detailed information: for each of the annotated MS/MS spectra presented in the main figures and figure S2 please indicate the name of the corresponding .RAW file and the scan number so that reinterpretation will be straightforward.

Figure 1:

☐ Figure 1a: please consider the comments above and enhance the sketch so that the general reader is immediately able to perceive the main advantages of the femtolaser system for biological systems.

☐ Figure 1b: It might be helpful to mention the DNA digesting enzymes in the legend. Besides, I have some minor concern regarding the combination of enzymes utilized. Benzonase is leading to digestion products ranging from mononucleotides (rare) to short DNA oligonucleotides (2 to 5 nucleotides) that all carry a 5'-monophosphate. Likewise, DNaseI produces oligonucleotides harboring a 5'-monophosphate. In contrast, MNase results in the generation of mononucleotides and short oligos possessing a 3'-phosphate. It is known for some RNases (but to my knowledge not for Benzonase and MNase) that they can also work as (ribo)nucleotide phosphatases. Hence the combination of Benzonase and MNase might reduce (in case they exhibit phosphatase activity) the amount of phosphorylated mononucleoside-peptide heteroconjugates thereby prohibiting their interaction with TiO. Do the authors have some indication that mononucleotide crosslinks are underrepresented in their data set? The use of C18 STAGE tips might lead to the loss of some peptide-DNA heteroconjugates (if the overall hydrophilicity is high). Please discuss this possibility (e.g. an additional thin layer of Perseptive OligoR3 material on top of the C18 sandwich might lead to an improvement). Importantly, the authors completely skip to consider that in case of "in vivo" samples the TiO beads will also retain other posttranslationally modified peptides like phosphopeptides. Did the authors assess in their data set obtained from the murine ES cell chromatin, if their workflow is also enriching phosphorylated or O-glycosylated peptides? Please discuss this.

Figure 2:

☒ Figure 2a: Maybe label Fig. 2a with “EMSA”. Which conditions (ratio between protein and DNA) were chosen for the experiments depicted in 2b-d? The EMSA lacks a specificity control in form of a double-stranded DNA oligo harboring a mutated NFI binding site. Alternatively, the binding reactions could be competed by an increasing amount of non-biotinylated DNA bearing a mutant NFI binding sequence.

☒ Figure 2b: Maybe label all immunoblots as “Western Blot”. Please provide the amount and concentration of DNA and protein. The authors claim that this figure unequivocally demonstrates the formation of UV crosslinked NFI DNA species that seem to migrate in two major forms in the denaturing SDS-PAGE. Since the read out of the gel is detecting the biotinylated DNA this is an over-interpretation of the data as long as DNA-DNA photo-crosslinking cannot be ruled out. This experiment has to be repeated with DNA bearing the WT NFI site and control DNA harboring a mutant NFI binding sequence. In addition, optional proteinase K treatment of some of the binding reactions/samples will clarify if the visualized gel bands change mobility in a protein-dependent manner. In this context the authors should also consider that the molecular weight of the putative crosslinked DNA-protein species differs in Fig. 2E (clear discrepancy!). In the latter case the read out of the gel bands is protein-centric (anti-His-tag antibody reaction with His-tagged NFI). Finally, the choice of total energies is not ideal for curve fitting and more measurement points should be provided, especially in light of the fact that the authors are claiming a two-photon process.

☒ Figure 2c: same comments and concerns as for Fig 2b.

☒ Figure 2d: Again, a series of control samples employing DNA containing a mutant NFI site are crucial in order to assess if the femtolaser crosslinking reaction can discriminate between sequence-specific and general DNA binding. Otherwise, the application of fLiX-MS to biological chromatin samples would face the problem of recovering functional and non-functional protein DNA interactions. Therefore, Reimer et al. should also discuss the potential utility of their method for interrogating DNA protein binding reactions conducted in the presence of cellular or nuclear extracts instead of using highly purified recombinant protein. As proposed for Fig 2b nuclease digested samples would provide additional evidence that the visualized gel bands are indeed covalently attached DNA-protein complexes. This would help to argue against certain literature reports (for example Itri, F. et al., 2016, Cell. Mol. Life Sci. 73, 637-648) that claim protein-protein crosslinking induced by UV laser irradiation.

In summary, without the proposed improvement of the data in Fig. 2 the conclusions drawn on the crosslinking efficiencies would have to be considered preliminary. Hence, it will be absolutely essential to improve the manuscript for this paragraph in order to be able to better judge the future impact and possible shortcomings of fLiX-MS.

Figure 3:

☐ Figure 3a: Please refer to comments stated below (concerning data presentation in the supplementary material). Please add an additional column, which provides information on the charge state, m/z value and the “mass accuracy” (mass deviation in ppm) of the parent ion.

☐ Figure 3a, b: and in general: Albeit the fact that the total number of identified crosslinked DNA-peptide spectra is limited, it would be nice if the authors could mention if they were able to spot preferential crosslinking of certain amino acid side chains?

☐ Figure 3c: I highly recommend discussing the expected but missed-out histone DNA interactions.

Overall a supplementary figure showing some standard assay(s) characterizing the quality of the in vitro assembled recombinant histone octamers would be helpful.

Figure 4:

☐ Figure 4a: Again, the charge state, the m/z value and the “mass accuracy” should be shown.

☐ Figure 4d and e: Normally one would assume that the formation of a covalent bond between an arginine side chain (or lysine side chain) and a bulky molecule like di- or tri-nucleotides would prevent the cleavage of this site by trypsin or LysC. Therefore I would suggest reconsidering the interpretation of the spectra. The $\gamma_{1+C'-A}$ ion in Fig. 4d is very weak and normally one would expect a strong γ_4 ion because of the preferential gas phase cleavage N-terminal to prolines. Maybe the crosslink is at a different position in the peptide, e.g. the internal arginine? Besides, the “ γ_7 ” is the singly charged (mono-protonated) peptide precursor ion that had lost DNA. The same holds true for “ γ_6 ” in Fig. 4e. There is also no need to consider that the crosslink is present at the C-terminal arginine residue in the spectrum in Fig. 4e.

☐ Figure 4f: Could the authors please specify the molecular identity (formula) of the trioxidized cysteine? I assume that they are referring to cysteic acid that is generated via sulfenic and sulfinic acid intermediates. Is this modification already present in the recombinant protein before UV irradiation or do Reim et al. have evidence that this is a photochemical process? In case of the latter please be reminded that photo oxidation of adenosine (oxidized A) would be isobaric to G. Hence

the interpretation of the crosslinked nucleotide mass shift could potentially be attributed to AoxT-HPO₃.

☐ Figure 4g: Again γ_{12} is equivalent to the peptide molecular ion (in this spectrum the doubly charged species) that had lost C. The proposed C⁻ ion exhibits indeed strong ion abundance. Nevertheless, in the RNA field, nucleobase ions are rarely observed in case of mononucleotide crosslinks, because the nucleobase is believed to be covalently bonded to an amino acid side chain. Hence, observation of this ion will only be feasible in case the bond could dissociate during HCD (e.g. an amide bond crosslink between the amino group of C and the aspartate side chain).

With the goal to ease data interpretation of expert readers the authors should provide (supplementary information) spectrum ion tables (as generated by DB search engines like Mascot, Protein Prospector or PEAKS studio) containing the theoretical fragment ions ((a-), b-, γ - and nucleobase shifted series; major expected neutral loss peaks) and highlighting the assigned peaks. The same applies to Figures 5, 6 and Supplementary Figure S2!

Figure 5:

☐ Figure 5a: Same comment as for Fig. 4a.

☐ Figure 5b: Fig. 5a lists three putative crosslinks at single amino acid resolution and two crosslinks at lower resolution but the chart just displays four of them.

☐ Fig. 5d: Honestly, I am unable to find any possibility for UV crosslinking the dinucleotide AC (or CA) in the sequence window depicted in Fig. 5b. There is only one AC (and one CA) a bit more downstream of the TATA box and one CA upstream of the TATA box (not shown in Fig. 5B; all present on the antisense strand) in the short synthetic oligo derived from the AdML promoter. Alternatively, Reim and colleagues try to propose the presence of two separately crosslinked mononucleotides {[A]+[C]}-HPO₃, namely A (from the sense strand) and C (antisense/reverse strand)? This discrepancy or confusion must be resolved during revision.

☐ Fig. 5e and g: First of all I would like to draw the attention to the low m/z range window of the MS/MS spectrum, which contains many strong peaks that are not assigned. Second, the generation of b₁ and therefore a₁ ions depends on the presence of “unusually” strong acylium ions that are - based on the classic work of Schlosser and Lehman - normally not observed in charge directed (mobile proton mediated) fragmentation pathways with the exception that a basic amino acid (K, R and H) is present at the peptide N-terminus (Hiserodt, R.D. et al., 2007, J. Am. Soc. Mass Spectrom. 18, 1414-1422) or the amino terminus is chemically modified (e.g. N-terminal acetylation). Probably, the authors are able to propose a hypothesis why the crosslinked C nucleobase could stabilize the b₁/a₁ ions. Third, even though I appreciate the clever solution for finding an interpretation of the MS₂ spectrum in Fig. 5e and g, I would like to mention that up to my knowledge such a mechanism

has never been proposed by the RNA field, or is this “plus CO” adduct an optional parameter in the newest version of RNPxl? I would also like to remind that mass spectrometry as such and without further help of chemistry experiments (e.g. chemical reactions carried out with the molecule of interest followed by MS analysis) is far from being a perfect technique for de-novo elucidation of chemical structures. Therefore, currently the data interpretation in Fig. 5e leads to a nice hypothesis that should be corroborated by repeating the experiment with a synthetic ds-DNA oligonucleotide harboring a [13C,15N]-labeled C at the suspected contact position (with TBP). The Munich area based company SILANTES should be able to provide this reagent. Alternatively, the utilization of recombinant SILAC-labeled (at arginines and lysines) TBP or tryptic digestion in the presence of [18O]-H₂O would at least serve as a prove for the identity of the γ_6 +CO to γ_8 +CO series as well as the peptide ion+CO peak assignment. Last, I would like to recommend changing the wording in the main text (line 286 to 288). Of course, the CO adduct is not present in the parent ion containing the C-HPO₃ crosslinked desoxyribonucleotide (heteroconjugate of LDLKTIALR and C-HPO₃). Therefore, the sentence is rather confusing.

Figure 6:

☐ Figure 6d: the authors could conduct an in depth proteome analysis of the chromatin input as well as check public repositories for murine ES cells (e.g. RNA-seq) with the goal to find out if all the possible proteins (Hes2, Oct1, Oct2, Oct11 and Mad2l2) are present in their chromatin sample or if they are even expressed.

☐ Figure 6e: Maybe reference this paper (Winter, D. et al., 2010, J. Am. Soc. Mass Spectrom. 21, 1814-1820) to explain the presence of the c1 ion.

Supplementary Figures:

☐ Supplementary Figure S2d: It might make more sense to assign the dinucleotide as [TA+HPO₃] because the internal fragments are shifted by a thymine adduct and there is a strong peak for the nucleobase A`.

☐ Supplementary Figure S3: there is one peptide ion (from HES-2, KPLLEL + [TT], MH33+) that exhibits a δm of 10.7 ppm even though the methods section claims that the DB search was performed with a maximum allowed mass deviation of +/- 10 ppm.

Generally speaking, it would be important to know if the MS1 spectra were recalibrated at the global level (as performed for instance by MaxQuant)? Since on one hand the PD version is not indicated in

the methods section and on the other hand I am not familiar with the details of PD data processing this is an important point because many spectra do actually show a δm of greater 5 ppm, which at least in my experience is unusual for correct IDs on data acquired on a HF-type orbitrap instrument, except recalibration was not carried out. In this context I would also like the authors to provide the δm values for the major fragment ions in the ion spectrum tables requested in my comments above (Figure 4). Information on the latter will provide additional trust in the manual curation of MS/MS spectra.

Minor points:

Title:

The utilized high resolution MS is nowadays standard and not worthwhile to be emphasized.

Figure 1:

□ Figure 1a: SHG (please explain the abbreviation in the legend)

Figure 2d:

□ Combining ant-His moAb with a fluorescence-labeled secondary reagent in conjunction with a fluorescence-labeled streptavidin followed by two color imaging would enable the simultaneous detection of both protein and DNA on the same blot. The subsequent overlay of the fluorescence colors would directly reveal co-migration of protein and DNA.

General: did the authors consider other search tools like the Xi search engine (recent work from the Rappsilber lab) or TagGraph (Nature Biotechnol. 37, 469-479)?

Reviewer #2 (Remarks to the Author):

Comments for the Authors

The manuscript by Reim et al. "Atomic-resolution mapping of transcription factor-DNA interactions by femtosecond laser crosslinking and high-resolution mass spectrometry" describes an innovative new method of analyzing proteins associated with DNA at specific locations and downstream applications. The method is elegant, and the workflow is clearly described.

The premise of this manuscript is simple: there are currently severe limitations on structural data as to how transcription factor (TF) bind their cognate DNA or nucleosomal templates in living cells. For example, ChIPseq reveals sequences that a given TF is associated with, but cannot distinguish whether the TF is binding directly to the DNA or via a complex of other DNA-binding TFs. In vitro experiments that involve recombinant proteins do not necessarily reflect in vivo results, and co-crystallization/NMR of protein-DNA interactions are not always possible, and when possible, may not reflect the native state of modified proteins/DNA/nucleosomes which bind a possibly modified complex of TFs.

To overcome this technical barrier to a complete understanding of how TFs behave, the authors have developed a novel approach based on xlinking mass spec. The method utilizes a high intensity UV femtosecond laser that can cross-link protein-DNA interactions, followed by digestion with nucleases and peptidases that cleave DNA/protein, followed by mass spec to discover the direct peptide-DNA interactions. This allows the interacting domains to be mapped directly, with higher resolution and accuracy, than more traditional means. To confirm, they used NF1 as an example and confirmed with this approach, and discovered interactions consistent with published biochemical literature. They also mapped additional sequences to the cytosines on the reverse TTGGA consensus sequence, a novel discovery not previously reported.

Overall, we thought this method/approach is novel and exciting, and will likely have widespread impact on research that relies heavily on crystallographic data that is incomplete/non-existent.

That said, the work is really a Methods manuscript than a research/discovery article and should be treated as such. Otherwise, authors should elaborate on how the additional novel interactions impacts transcription and for which genes, which would require additional experiments such as ChIPseq and qRT-PCR to verify that their findings influence transcription in vivo (which we imagine they would prefer to avoid). If rewritten as a novel methods paper. this manuscript will be of interest to a wide readership and is suitable for the journal.

We have two suggestions for improvement before acceptance:

1) The authors state they used in vitro reconstituted nucleosomes, but figures and methods describing how this was done are missing- for this kind of novel methodology, please provide these. This suggestion does not impact the central findings of the paper, but will increase confidence in the interactome deduced from the ms/ms data.

2) The authors used TiO₂ for the enrichment of the crosslinked peptides and subsequently eliminated all the other remaining peptides in their data analysis step. However, TiO₂ enriches many of the other types of modified peptides (phosphopeptides, ADP-ribosylated peptides) as well as acidic peptides, which may be ignored during the data analysis, but they are present in the spectra, therefore possibly suppressing the signal of the crosslinked peptides or skewing the quantification. Please comment on this point.

-This report was written by Drs. Yamini Dalal, Aleksandra Nita-Lazar and Minh Bui at the National Institutes of Health in Bethesda.

Reviewer #3 (Remarks to the Author):

This is a paper that builds upon a body of prior literature, dating back around 10-20 years by some of the authors, which explored the use of femtosecond laser irradiation for DNA protein crosslinking. In the description of the laser-based method, the paper is reminiscent of a recent study from the Altucci group ("Time resolved analysis of DNA-protein interactions in living cells by UV laser pulses", *Scientific Reports*, 7, 2017), who also explored the use of femtosecond lasers for DNA-protein crosslinking in situ (including transcription factors and histones). They didn't use mass spec, which is essentially the major innovation of the current paper. Normally this wouldn't be enough in the way of novelty, but "solving" the detectability problem for protein-DNA linkages has been a challenge, and so worthy of consideration if the solution is a durable one. There are some technical matters that dampen my enthusiasm, however. It is hard to get a good sense of how useful a development has been presented.

I have the following concerns:

1. The authors should be using scrambled oligos as a control wherever possible. As it stands, all we have to convince us that the interactions are legitimate is agreement with other studies. This is useful, but not a replacement for controls. In this context, dismissing the unexpected crosslinks (based on distance from the canonical binding sites) as being due to conformational change is an unsupported claim. It can also be due to diffusion of the activated species to sites of high reactivity, and thus not reflective the ensemble of conformers at all. A lot can happen during ~50 seconds of irradiation!

2. There are other claims with weaker support than I think is justified, in the form of the identification of binding sites on the oligos. For example, what is the evidence that the cytosine crosslinks are specific for the reverse strand in the NF1 experiment (there are C's elsewhere) . Another unsupported claim is discounting Mad2I2 as the binding site for peptide KPLLEK, simply because it doesn't fit with expectations for DNA binding.

3. What are the yields of crosslinked products? It would be useful to know how robust of a method this is from that standpoint, as the numbers of detected crosslinks are not particularly high. Is this because of LC-MS issues or the enrichment methodology?

4. Have all the modification types been considered? The software used was primarily designed for RNA-protein interactions.

5. Page 8, line 159. You cannot avoid DNA damage at these fluences. Minimize perhaps, but not avoid. You will also get protein-protein crosslinking between aromatic amino acids. These issues should be clarified.

6. The identification of crosslinked sites is carefully done and reflective of the quality of the software used. There are a couple of puzzles though.

Line 244: is H₂O₂ loss of two OH's? Or one water and an H abstraction?

Line 292: the evidence that both L1789 and K181 are crosslinked at the same time to the same cytosine is unclear. Could these not be chimeric spectra? It would be useful to convey the peptide retention times for this (and the other features). Based on the chemical scheme presented in figure 5g, I find double coupling unconvincing.

7. The discussion section adds very little to the paper, as it is really a restating of the results section, and the claims in the final paragraph seem a bit overstated.

Response letter for Reim et al. - Atomic-resolution mapping of transcription factor-DNA interactions by femtosecond laser crosslinking and high-resolution mass spectrometry

We thank the reviewers for their constructive comments, which we have addressed in the revised manuscript. We believe that our manuscript greatly profited from the reviewer’s feedback. Below is our point-to-point response to the raised topics.

Reviewer #1 (Remarks to the Author):

Overview:

More than thirty years ago pioneering studies by the J.T. Lis laboratory in *E. coli* and *Drosophila* (Gilmour and Lis, 1984, PNAS 81; Gilmour, D.S. et al, 1986, Cell 44) demonstrated the potential power of UV irradiation mediated protein DNA crosslinking for studying the biology of DNA binding proteins. As a matter of fact these papers are also the first ones describing a kind of chromatin IP assay, which is nowadays in most of the cases carried out employing formaldehyde (FA) crosslinking. Follow-up work by the laboratories of M.D. Biggin, D. Houde and G.L. Hager (Walter, J. and Biggins, M.D., 1997, Methods 11; Moss, T. et al., 1997, Methods 11; Nagaich, A.K. et al., 2004, Mol. Cell 14) demonstrated that the efficiency of UV DNA protein crosslinking can be increased by utilizing pulsed UV light emitted from lasers. Concomitantly, literature reports from the last 20 years also suggest that the technological evolution of laser technology (e.g. the move from nanosecond to picosecond laser pulsing technology) bear the potential to address the major shortcoming of UV mediated DNA protein crosslinking, namely its low efficiency that comes along with extremely low recovery of covalently bound DNA protein complexes hampering the general applicability of the approach.

This is in stark contrast to the field of RNA protein interactions for which UV crosslinking has been and still is the mainstay of any genomics or proteomics follow-up methodology. Until now, the huge difference in the yield of UV-crosslinked RNA-protein versus DNA-protein complexes is explained by the compromised photo activation of the nucleobases in the double stranded DNA (base pairing, base stacking) versus the situation found in single stranded RNA. Counteracting this by prolonging the exposure time (using standard UV lamps) is not feasible because under such conditions severe photo damage of biological macromolecules prohibits subsequent biochemical workflows (e.g. cells cannot be efficiently lysed anymore).

Therefore, FA crosslinking, which exhibits recovery of DNA-bound proteins in the range between 0.01-2.0% and that forms metastable zero-length covalent DNA-protein complexes (but is partially also recovering piggy-back/indirect DNA-protein interactions) is currently the commonly employed starting point of any large scale experiments like ChIP-seq, FAIRE-seq and Hi-C/3C-seq.

In this context the proposed manuscript by Reim and colleagues tries to overcome the current bottlenecks of examining DNA-protein interactions with the help of UV irradiation by introducing the application of femtolaser pulse technology in conjunction with state-of-the-art nanoLC-MS analysis of DNA-crosslinked tryptic peptides. At first the authors claim extremely

high crosslinking efficiency (up to 100%) in reconstituted in vitro systems but the efficiencies of crosslinking in live cells as well as quantifying the ability to recover the UV XLinked molecules are not addressed.

Nevertheless, this is with no doubt a very interesting and pioneering study which should be published in a decent journal like Nature Communications. Emphasizing the challenging task of mapping the interaction surface between proteins and DNA, this is much needed for the field. The used approach is innovative, appropriate and a step in the right direction but for reasons outlined below a couple of issues mainly related to (quality) control experiments, data representation, the need for a much better and detailed description of the methods and a more critical discussion have to be resolved in a major revision prior to publication of the work.

In addition and since this is important for the editor to judge future citation of this study, the general availability and robustness (easy to replicate?) of the experimental femtolaser setup is currently not clear at all. In other words, could Wierer, Reim and coworkers envision that their technology will become available on a routine basis in the near future?

Therefore, I would like to recommend the work of Reim and colleagues for publication in Nature Communications given that the important concerns rose above and detailed below can be rectified in form of a major revision.

We are pleased by the positive and constructive feedback of reviewer #1 and especially thank him or her for placing our work in a historical context.

Major points:

Description of methods: the materials and methods section is lacking information that is crucial for both understanding the paper and the possibility to reproduce the work.

– In case of the synthetic oligonucleotides it is not clear if the 5´-biotinylation is only present at the “sense” strand or on both (also reverse) oligos?

The 5´-biotinylation is present on both forward and reverse oligo. We have included this information in the new material and methods section.

– In general, it is difficult to extract information concerning the concentration and total amount of protein(s) and DNA in the reconstituted in vitro systems (NFI, TBP, reconstituted nucleosome core particles)

We have expanded the information regarding the exact amounts and concentrations in the material and methods section.

– The LC/MS methods do not provide the following parameter settings: source temperature, RF-value for S lense (or equivalent of it), target value and injection time for MS1

We have improved the description of LC/MS methods including the missing information

– The description of the data analysis is lacking the following parameter description: number of max. allowed missed cleavages, PD and RNPxl version and the criteria applied for manual curation (e.g. what is the percentage of spectra initially proposed by PD/RNPxl that had been manually removed?)

We now include more detailed information on maximally allowed missed cleavages, version information for Proteome Discoverer and RNPxl as well as a detailed description of the criteria for manual spectra curation in the material and methods section. Due to the very stringent filter criteria the number of spectra that remain after filtering the list of initially proposed cross-links by RNP-XL is < 1% of the original list. Therefore, we do not use RNPxl as full analysis tool, but rather as an initial filter for spectra identification. We believe that a future software, which can automatically detect DNA-crosslinked peptides based on the criteria described in our manuscript, would be a highly valuable tool, and we are currently envisioning such a development as an add on to the new AlphaPept software that will be published in a few months.

– The sketch of the femtolaser setup (see Fig. 1) and its description makes it complicated to compare the work presented by Reim et al. with the literature (picolaser or nanolaser configurations etc.). Please indicate the pulse repetition rate (is it 0.5 MHz?), the origin/manufacturer of the b-barium borate crystal and the configuration of the active fiber systems laser (“high-repetition rate” or “high pulse energy”). It would be very beneficial for the general reader if the authors could provide in the main text an idea on how much average power (Watt) and peak power (Watt) is actually applied to the samples (also W/cm²). Moreover, it would be nice to clarify (for their standard parameters of 500 fs pulses at 50 nJ and a total energy of 1.25 J) how many pulses (equal to the total exposure time of the biological samples) are actually required (about 5,000? ~10 ms exposure time?). I am asking this because until now it seems that there is no standard in presenting such data and older literature is often using W and W/cm² in the figures.

We have greatly extended the description of the laser setup in the material and methods section including all variables requested by reviewer #1 and updated figure 1 as suggested.

Finally, estimation on how many photons per nucleotide are actually needed to reach saturating photo crosslinking of NFI to the synthetic DNA would be quite nice to enhance the understanding of the work by non-physicists.

Quantum efficiencies in general and the number of photons per nucleotide in our case are very helpful parameters to quantify classical photochemical processes based on the absorption of single photons. Several particularities complicate the classical photochemical description of the laser-induced processes, which are described in the publication. These are:

Femtochemistry: Due to the high intensities used in the femtosecond range and the underlying 2-photon processes, non-linear intensity dependence of the absorption processes occur. Moreover, the absorption cross section is still unknown.

Biological model: The paper uses ‘xlinked species’ as parameter to quantify xlink yield (Figure 2). Xlinked species describes the covalent binding of the transcription factor to the DNA and can be achieved by at least one xlink of a nucleotide to the amino acids in a DNA-Binding sequence. We have identified five crosslinks locations on the palindromic consensus DNA-binding sequence of NF1 exist (see Figure 4c), which can potentially contribute alone or together with others to the parameter ‘xlinked species’.

Therefore, we believe that due to the complex, underlying physical processes with non-linear dependencies and the chosen method for determining the cross-linking efficiency, an estimation of the quantum efficiency is afflicted with too high inaccuracies and requires too many assumptions.

Data availability: the data are uploaded to the EBI PRIDE data base but unfortunately no reviewer login credentials have been communicated. Please allow the reviewers to access the .RAW files during the revision process. Additionally, it would be helpful to provide the following detailed information: for each of the annotated MS/MS spectra presented in the main figures and figure S2 please indicate the name of the corresponding .RAW file and the scan number so that reinterpretation will be straightforward.

This was provided already but we apologized if it was not in an obvious place: The PRIDE login information is as follows: Project accession: PXD014898, Username: reviewer75731@ebi.ac.uk, Password: VVhUy6lh

We have now included the raw file name and scan number relationships of all presented spectra in Supplemental table 1.

Figure 1:

Figure 1a: please consider the comments above and enhance the sketch so that the general reader is immediately able to perceive the main advantages of the femtolaser system for biological systems.

We have updated Figure 1 to make it more conceivable to the general reader.

Figure 1b: It might be helpful to mention the DNA digesting enzymes in the legend.

We have included the information on digesting enzymes in the Figure legend.

Besides, I have some minor concern regarding the combination of enzymes utilized. Benzonase is leading to digestion products ranging from mononucleotides (rare) to short DNA oligonucleotides (2 to 5 nucleotides) that all carry a 5′-monophosphate. Likewise, DNase I produces oligonucleotides harboring a 5′-monophosphate. In contrast, MNase results in the generation of mononucleotides and short oligos possessing a 3′-phosphate. It is known for some RNases (but to my knowledge not for Benzonase and MNase) that they can also work as (ribo)nucleotide phosphatases. Hence the combination of Benzonase and MNase might reduce (in case they exhibit phosphatase activity) the amount of phosphorylated mononucleoside-peptide heteroconjugates thereby prohibiting their interaction with TiO. Do the authors have some indication that mononucleotide crosslinks are underrepresented in their data set? The use of C18 STAGE tips might lead to the loss of some peptide-DNA heteroconjugates (if the overall hydrophilicity is high). Please discuss this possibility (e.g. an additional thin layer of Perseptive OligoR3 material on top of the C18 sandwich might lead to an improvement). Importantly, the authors completely skip to consider that in case of “in vivo” samples the TiO beads will also retain other posttranslationally modified peptides like phosphopeptides. Did the authors assess in their data set obtained from the murine ES cell chromatin, if their workflow is also enriching phosphorylated or O-glycosylated peptides? Please discuss this.

These are very interesting notes. We have intensively queried pubmed and other sources to find out whether MNase or Benzonase might possess phosphatase activity, and did not find a published evidence for this. Although we cannot exclude it completely, we believe that there is not such activity. About 35% of all crosslinks identified in this study are mononucleotides, suggesting that our method is capable of capturing such species.

The use of OligoR3 is an interesting proposal, which we will investigate in future. As RNA-crosslinks and hydrophilic phosphopeptides are commonly purified by C18 material (for instance see Kramer et al. 2014, Nature Methods), we believe that this purification method would be well suited for DNA as well, although we are very motivated in testing alternatives. In addition to OligoR3 we could also imagine graphite columns as an alternative, which are known to favor hydrophilic peptides over non-hydrophilic ones (Larsen et al. 2004, Mol Cell Proteomics).

We purified our “in-vivo” samples in multiple steps that excluded the enrichment of phosphorylated peptides. These involved (i) biotinylation of digested chromatin, (ii) boiling under denaturing conditions, (iii) enrichment and purification of biotinylated DNA under denaturing conditions, (iv) elution of DNA crosslinked proteins by DNA digestion and (v) protein digestion and (vi) TiO₂ enrichment of crosslinked peptides. While phosphorylated peptides of non-DNA-crosslinked proteins are removed in step (iii), only phosphorylated peptides of DNA crosslinked peptides would theoretically be enriched. In fact, performing an open search in pFind (the search engine which to our experience reaches the highest depth of identifying post translationally modified peptides), indicates that only 3.4% of all enriched unique peptides are phosphorylated on serine or aspartic acid and no other phosphorylation type. All phosphorylated peptides were derived from three proteins - Hsp90b, Dcaf8 and Pairb - which can all bind directly to chromatin.

Figure 2:

Figure 2a: Maybe label Fig. 2a with “EMSA”. Which conditions (ratio between protein and DNA) were chosen for the experiments depicted in 2b-d? The EMSA lacks a specificity control in form of a double-stranded DNA oligo harboring a mutated NFI binding site. Alternatively, the binding reactions could be competed by an increasing amount of non-biotinylated DNA bearing a mutant NFI binding sequence.

We have marked the blot as “EMSA” and included detailed information on amount and molar ratios in the material and methods sections.

We have also repeated the experiment including DNA comparing wild type and mutated NFI binding site. Please note that NF1 also has unspecific binding (Dekker et al. 1996, MCB), which likely explains the faint shifted band for the mutated DNA oligo.

Figure 2b: Maybe label all immunoblots as “Western Blot”.

We followed the recommendation for labeling the blots as suggested.

Please provide the amount and concentration of DNA and protein.

We now include this information in the material and methods section.

The authors claim that this figure unequivocally demonstrates the formation of UV crosslinked NFI DNA species that seem to migrate in two major forms in the denaturing SDS-PAGE. Since the read out of the gel is detecting the biotinylated DNA this is an over-interpretation of the data as long as DNA-DNA photo-crosslinking cannot be ruled out. This experiment has to be repeated with DNA bearing the WT NFI site and control DNA harboring a mutant NFI binding sequence. In addition, optional proteinase K treatment of some of the binding reactions/samples will clarify if the visualized gel bands change mobility in a protein-dependent manner. In this context the authors should also consider that the molecular weight of the putative crosslinked DNA-protein species differs in Fig. 2E (clear discrepancy!). In the latter case the read out of the gel bands is protein-centric (anti-His-tag antibody reaction with His-tagged NFI).

We appreciate the comments and added additional controls for this part. First, we repeated the total energy curve experiment using DNA oligos harboring either a NF1 consensus sequence, or a mutated form of it (Figure 2d). This showed that the signal in the DNA-centric Western-blots is clearly dependent on specific interaction of NF1 with DNA. Second, to clarify the discrepancies in molecular weight, we performed a Western Blot for the UV crosslinked NF1-DNA complex after digest with or without prior proteinase K or DNase I treatment (Figs. 2e, S1d). After detection of the His-tagged NF1, we stripped the membrane and probed it with the

Streptavidin-HRP conjugate to detect biotin-labeled DNA. We realized that about 7.5% of NF1 crosslinked to single- or double stranded DNA under these conditions, while a major part (53%) shifted to higher molecular weight level, indicating that protein-protein crosslinking does take place (Fig. S1). In fact, the high molecular weight band at ~115kDa corresponds to two NF1 molecules, while the two lower bands fit to the molecular weight of NF1 bound to a single or double-stranded NF1 oligo. Notably, the signal of the high-molecular weight fraction disappeared with DNase I digestion, indicating that the signal represents protein-DNA crosslinks. While the mono-NF1-DNA crosslinks disappeared with proteinase K treatment, the crosslinked bands in the high molecular weight region are not well enough resolved, to allow differentiation between protein-protein and protein-protein-DNA crosslinks. Extrapolating from the crosslinking efficiency of mono-NF1-DNA and the intensities of the 65 of kDa, 115kDa and 185kDa bands in the DNA-biotin blot, we estimate a crosslinking efficiency of 14% under the given energy conditions.

To improve readability, we decided to show only the bands corresponding to the mono NF1-DNA crosslinked complex throughout Fig. 2, and provide the full blots as supplemental data in Supplemental Figure S1.

Finally, the choice of total energies is not ideal for curve fitting and more measurement points should be provided, especially in light of the fact that the authors are claiming a two-photon process.

Our new energy curve (Figure 2d) includes more data points and shows that saturation is reached already at about 316 mJ. With increasing total energy the generation of higher molecular weight species predominates leading to a decrease in intensity of the mono NF1-DNA band.

Figure 2c: same comments and concerns as for Fig 2b.

Please see the answers just above.

Figure 2d: Again, a series of control samples employing DNA containing a mutant NFI site are crucial in order to assess if the femtolaser crosslinking reaction can discriminate between sequence-specific and general DNA binding. Otherwise, the application of fliX-MS to biological chromatin samples would face the problem of recovering functional and non-functional protein DNA interactions.

Please see answers above just above.

Therefore, Reim et al. should also discuss the potential utility of their method for interrogating DNA protein binding reactions conducted in the presence of cellular or nuclear extracts instead of using highly purified recombinant protein.

Although we thought about the possibility to study protein-DNA binding with nuclear extracts, we decided that UV-crosslinking of cells would be a better way to show that the method is capable to capture more ‘in vivo’ protein-DNA interaction events, given its fully unbiased character. Nevertheless, we believe that the proposed experiment should be feasible for clarifying DNA interactions for proteins that have a known and specific consensus site, but are difficult to purify and we now include the potential utility of this in the discussion section.

As proposed for Fig 2b nuclease digested samples would provide additional evidence that the visualized gel bands are indeed covalently attached DNA-protein complexes. This would help to argue against certain literature reports (for example Itri, F. et al., 2016, Cell. Mol. Life Sci. 73, 637-648) that claim protein-protein crosslinking induced by UV laser irradiation.

Please see answers above.

In summary, without the proposed improvement of the data in Fig. 2 the conclusions drawn on the crosslinking efficiencies would have to be considered preliminary. Hence, it will be absolutely essential to improve the manuscript for this paragraph in order to be able to better judge the future impact and possible shortcomings of fliX-MS.

Integrating the suggestions of reviewers #1 and #3 we believe that this part of our paper strongly improved from them.

Figure 3:

Figure 3a: Please refer to comments stated below (concerning data presentation in the supplementary material). Please add an additional column, which provides information on the charge state, m/z value and the “mass accuracy” (mass deviation in ppm) of the parent ion.

We have included information on charge state, m/z value and mass accuracy deviation in the table presented in Fig. 3a.

Figure 3a, b: and in general: Albeit the fact that the total number of identified crosslinked DNA-peptide spectra is limited, it would be nice if the authors could mention if they were able to spot preferential crosslinking of certain amino acid side chains?

We analyzed preferential crosslinking among 14 DNA-crosslink peptides, where the crosslink unequivocally mapped to a single amino acid. While we identified an underrepresentation of amino acids with aromatic side chains (top panel), this effect disappeared, when normalizing on the amino acid frequencies reported by the most recent Uniprot release (2019_11) (bottom panel). Most notably, we observed a complete absence of DNA crosslinks on acidic side chains,

however, given the overall low statistical power of this analysis, we decided to not include it in the manuscript.

Figure 3c: I highly recommend discussing the expected but missed-out histone DNA interactions.

One limitation of our current assay format is the use of trypsin for protein digestion, restricting the potential detection to tryptic peptides. Due to the high percentage of lysine and arginine residues, histones are particularly challenging for proteomic analyses. Together with effects derived from different crosslink efficiencies, this contributes to the non-complete coverage of

histone DNA interactions. We included an explanation and possible solutions in the discussion section.

Overall a supplementary figure showing some standard assay(s) characterizing the quality of the in vitro assembled recombinant histone octamers would be helpful.

We have included a Coomassie stained SDS-PAGE and DNA retardation gel comparing free and nucleosome bound DNA in Figure S2.

Figure 4:

Figure 4a: Again, the charge state, the m/z value and the “mass accuracy” should be shown.

We have included the missing information in the table presented in Fig. 4a.

Figure 4d and e: Normally one would assume that the formation of a covalent bond between an arginine side chain (or lysine side chain) and a bulky molecule like di- or tri-nucleotides would prevent the cleavage of this site by trypsin or LysC. Therefore I would suggest reconsidering the interpretation of the spectra. The $y1+C'-A$ ion in Fig. 4d is very weak and normally one would expect a strong $y4$ ion because of the preferential gas phase cleavage N-terminal to prolines. Maybe the crosslink is at a different position in the peptide, e.g. the internal arginine? Besides, the “ $y7$ ” is the singly charged (mono-protonated) peptide precursor ion that had lost DNA. The same holds true for “ $y6$ ” in Fig. 4e. There is also no need to consider that the crosslink is present at the C-terminal arginine residue in the spectrum in Fig. 4e.

We agree with reviewer #1 that a crosslink on the terminal arginine would actually interfere with trypsin digestion. In fact, as the intensity of the $y1^C$ ion is very low, and as it represented the only evidence for a crosslink location on R89, we removed the localization annotation for this peptide. Likewise, we agree that the crosslink localization on the terminal arginine in DCP15 (Fig. 4e) is unlikely as well. As in this case the crosslink information is based on the presence of an arginine immonium ion, we were able to locate the crosslink on R117 instead. We have updated Fig. 4 and the text accordingly.

Figure 4f: Could the authors please specify the molecular identity (formula) of the trioxidized cysteine? I assume that they are referring to cysteic acid that is generated via sulfenic and sulfinic acid intermediates.

The trioxidation we refer to is indeed cysteic acid (Williams et al. 2011, J Am Soc Mass Spectrom). We clarified this in the text.

Is this modification already present in the recombinant protein before UV irradiation or do Reim et al. have evidence that this is a photochemical process? In case of the latter please be reminded that photo oxidation of adenosine (oxidized A) would be isobaric to G. Hence the

interpretation of the crosslinked nucleotide mass shift could potentially be attributed to AoxT-HPO₃.

Although we cannot exclude that oxidation might occur during UV-radiation, we believe that cysteine trioxidation occurs during sample preparation similar to oxidation of methionine. Standard proteomic workflows include reduction and alkylation of cysteine residues (which would remove the modification). Here, we chose not to include these steps to avoid losing potential crosslinks that would be reduced by DTT or introducing alkylation artifacts. In fact, trioxidation of cysteine has been reported as a common side reaction in standard buffer conditions during MS sample preparation in absence of reduction / alkylation (Bayer and König, 2016, Rapid Commun. Mass Spectrom.). In addition, revisiting our data we identified cysteic acid containing peptides also in the control samples that were not UV-irradiated. We therefore changed the text accordingly.

Figure 4g: Again y₁₂ is equivalent to the peptide molecular ion (in this spectrum the doubly charged species) that had lost C. The proposed C⁻ ion exhibits indeed strong ion abundance. Nevertheless, in the RNA field, nucleobase ions are rarely observed in case of mononucleotide crosslinks, because the nucleobase is believed to be covalently bonded to an amino acid side chain. Hence, observation of this ion will only be feasible in case the bond could dissociate during HCD (e.g. an amide bond crosslink between the amino group of C and the aspartate side chain).

While the free nucleobase could indicate a crosslink via the deoxyribose part, the existence of a nucleobase crosslink on the y₃ ion clearly shows that the crosslink is nucleobase directed. Hence, we believe that the free nucleobase must indeed be dissociated from the crosslinked peptide during HCD fragmentation.

With the goal to ease data interpretation of expert readers the authors should provide (supplementary information) spectrum ion tables (as generated by DB search engines like Mascot, Protein Prospector or PEAKS studio) containing the theoretical fragment ions ((a-), b-, y- and nucleobase shifted series; major expected neutral loss peaks) and highlighting the assigned peaks. The same applies to Figures 5, 6 and Supplementary Figure S2!

This is a very good point and we have now included detailed tables with theoretical and observed fragment ions along with observed mass shifts and mass accuracies for all reported MS₂ spectra as Supplemental file 1.

Figure 5:

Figure 5a: Same comment as for Fig. 4a.

We have included information on charge state, m/z value and mass accuracy in the table presented in Fig. 5a.

Figure 5b: Fig. 5a lists three putative crosslinks at single amino acid resolution and two crosslinks at lower resolution but the chart just displays four of them.

This discrepancy is due to the proposed double crosslink of the LDLKTIALR peptide to cytosine.

Fig. 5d: Honestly, I am unable to find any possibility for UV crosslinking the dinucleotide AC (or CA) in the sequence window depicted in Fig. 5b. There is only one AC (and one CA) a bit more downstream of the TATA box and one CA upstream of the TATA box (not shown in Fig. 5B; all present on the antisense strand) in the short synthetic oligo derived from the AdML promoter. Alternatively, Reim and colleagues try to propose the presence of two separately crosslinked mononucleotides {[A]+[C]}-HPO₃, namely A (from the sense strand) and C (antisense/reverse strand)? This discrepancy or confusion must be resolved during revision.

We agree with reviewer #1 that the annotation of this crosslinked peptide was not clear and have now improved it. Our data indicates two independent crosslinks to A and C on the same peptide, which are evidenced by 3 and 4 unique shifted product ions, respectively.

Fig. 5e and g: First of all I would like to draw the attention to the low m/z range window of the MS/MS spectrum, which contains many strong peaks that are not assigned.

The peaks in the low m/z range are sub-peptide size and therefore likely fragmentation products, which are difficult to annotate. Individual analysis of these peaks identified one as the immonium ion of lysine, and another one as deoxyribose after loss of CO (see below).

Second, the generation of b1 and therefore a1 ions depends on the presence of “unusually” strong acylium ions that are - based on the classic work of Schlosser and Lehman - normally not observed in charge directed (mobile proton mediated) fragmentation pathways with the exception that a basic amino acid (K, R and H) is present at the peptide N-terminus (Hiserodt, R.D. et al., 2007, J. Am. Soc. Mass Spectrom. 18, 1414-1422) or the amino terminus is chemically modified (e.g. N-terminal acetylation). Probably, the authors are able to propose a hypothesis why the crosslinked C nucleobase could stabilize the b1/a1 ions.

This is a very interesting note. However, we believe that the discrepancy results from the use of HCD fragmentation in contrast to low-energy CID fragmentation that was applied in the above mentioned work. A more recent study from the Yates group (Diedrich et al. 2013, J Am Soc Mass Spectrom) studied the fragmentation series of peptides in response to increasing HCD energy observing a strong increase of smaller a and b ions with increased energy. In fact one of the peptides in the study (LTILLEELR) contained an N-terminal leucine similar to DCP20. Starting at a normalized collision energy of 20, the authors observed a strong presence of a leucine immonium ion, which is indistinguishable from the a1 ion.

Another study from our group (Michalski et al. 2012, JPR) also systematically investigated HCD fragmentation spectra and reported an a1 ion representing a phosphorylated N-terminal serine. We therefore believe that the deoxycytidine crosslinked to the a1 ion should be equally possible to be generated by high-energy HCD fragmentation.

Third, even though I appreciate the clever solution for finding an interpretation of the MS2 spectrum in Fig. 5e and g, I would like to mention that up to my knowledge such a mechanism has never been proposed by the RNA field, or is this “plus CO” adduct an optional parameter in the newest version of RNPxl? I would also like to remind that mass spectrometry as such and without further help of chemistry experiments (e.g. chemical reactions carried out with the molecule of interest followed by MS analysis) is far from being a perfect technique for de-novo elucidation of chemical structures. Therefore, currently the data interpretation in Fig. 5e leads to a nice hypothesis that should be corroborated by repeating the experiment with a synthetic ds-DNA oligonucleotide harboring a [13C,15N]-labeled C at the suspected contact position (with TBP). The Munich area based company SILANTES should be able to provide this reagent. Alternatively, the utilization of recombinant SILAC-labeled (at arginines and lysines) TBP or tryptic digestion in the presence of [18O]-H2O would at least serve as a prove for the identity of the y6+CO to y8+CO series as well as the peptide ion+CO peak assignment.

We followed the suggestions of reviewer #1 and ordered an oligonucleotide, in which the first C on the commentary strand downstream of the TATAAAA sequence was isotopically labeled by 13C and 15N. We repeated the experiment using this oligo, however due to technical problems on the mass spectrometry side we were not able to analyze this experiment and we were not able to access the laser in the time-frame of the revision for a repetition of the experiment (also due to the initial long synthesis time of the oligo).

However, revisiting the data, especially the peaks of the lower mass range, yielded an interesting observation, which points to an alternative model for generation of the CO-crosslinked product ions. The most intense peak in the low m/z range (m/z = 89.06) equals deoxyribose after loss of CO, which can be seen as a fragment ion of the deoxycytidine -CO ion, after the loss of cytosine. This ion provides evidence that the CO adduct on the product ions results from a fragmented deoxyribose, rather than being the product of cytosine fragmentation.

Taken together, our data indicates that one crosslink targets the ribose part of the deoxycytidine while the other crosslink targets the nucleobase. We therefore changed the proposed model in Figure 5g accordingly and updated the text.

Last, I would like to recommend changing the wording in the main text (line 286 to 288). Of course, the CO adduct is not present in the parent ion containing the C-HPO3 crosslinked desoxyribonucleotide (heteroconjugate of LDLKTIALR and C-HPO3). Therefore, the sentence is rather confusing.

We agree that the sentence was not very clear and removed it.

Figure 6:

Figure 6d: the authors could conduct an in depth proteome analysis of the chromatin input as well as check public repositories for murine ES cells (e.g. RNA-seq) with the goal to find out if all the possible proteins (Hes2, Oct1, Oct2, Oct11 and Mad2l2) are present in their chromatin sample or if they are even expressed.

We appreciate this suggestion and included a deep proteomic measurement (> 9700 proteins) of the same ES cell line (Supplemental Figure S4b) than the one used for fliX-MS. The exclusive presence Oct1 and Oct11 in this dataset suggests that the identified crosslink derives from either of these two transcription factors.

Figure 6e: Maybe reference this paper (Winter, D. et al., 2010, J. Am. Soc. Mass Spectrom. 21, 1814-1820) to explain the presence of the c1 ion.

We followed the suggestion of reviewer #1 and included the reference in the manuscript.

Supplementary Figures:

Supplementary Figure S2d: It might make more sense to assign the dinucleotide as [TA+HPO₃] because the internal fragments are shifted by a thymine adduct and there is a strong peak for the nucleobase A`.

We have modified the figure accordingly.

Supplementary Figure S3: there is one peptide ion (from HES-2, KPLLEL + [TT], MH33+) that exhibits a Δm of 10.7 ppm even though the methods section claims that the DB search was performed with a maximum allowed mass deviation of +/- 10 ppm.

This peptide was identified by RNP-XL as an AA-H₂O crosslink. However, as the spectra contained a marker ion for thymine, and the mass of the AA-H₂O adduct is very close to TT (626.1152 vs. 626.1027), we manually annotated the peptide as a TT crosslink, even if the mass deviation slightly exceeded the 10 ppm limit.

Generally speaking, it would be important to know if the MS1 spectra were recalibrated at the global level (as performed for instance by MaxQuant)? Since on one hand the PD version is not indicated in the methods section and on the other hand I am not familiar with the details of PD data processing this is an important point because many spectra do actually show a Δm of greater 5 ppm, which at least in my experience is unusual for correct IDs on data acquired on a HF-type orbitrap instrument, except recalibration was not carried out. In this context I would also like the authors to provide the Δm values for the major fragment ions in the ion spectrum tables requested in my comments above (Figure 4). Information on the latter will provide

additional trust in the manual curation of MS/MS spectra.

To our knowledge and based on the documentation, the RNP-XL workflow does not contain a recalibration step. We have followed the suggestions of reviewer #1 and included Δm values in Supplemental table 1 containing all theoretical and identified fragment ions.

Minor points:

Title:

The utilized high resolution MS is nowadays standard and not worthwhile to be emphasized.

We have remove the phrase ‘high-resolution’ from the title.

Figure 1:

Figure 1a: SHG (please explain the abbreviation in the legend)

We have replaced SHG (second harmonic generator) by ‘borate crystal’ and explained its function in the figure.

Figure 2d:

Combining ant-His moAb with a fluorescence-labeled secondary reagent in conjunction with a fluorescence-labeled streptavidin followed by two color imaging would enable the simultaneous detection of both protein and DNA on the same blot. The subsequent overlay of the fluorescence colors would directly reveal co-migration of protein and DNA.

This is a good idea, however, as we did not have access to an imager that can detect fluorescence different from SYBR green, we decided for stripping the western blot and reblotting with the streptavidin-HRP probe (Figure 2e). As both the band in the control lane and the signal in the lane with the DNase treated sample disappeared, we are confident that we did not have any signal carry-over from the initial western-blot. In both cases the band corresponding to the crosslinked NF1 protein was detected at the same height.

General: did the authors consider other search tools like the Xi search engine (recent work from the Rappsilber lab) or TagGraph (Nature Biotechnol. 37, 469-479)?

We did not consider these two search engines for this study, but we will look into them in future.

Reviewer #2 (Remarks to the Author):

Comments for the Authors

The manuscript by Reim et al. “Atomic-resolution mapping of transcription factor-DNA interactions by femtosecond laser crosslinking and high-resolution mass spectrometry” describes an innovative new method of analyzing proteins associated with DNA at specific locations and downstream applications. The method is elegant, and the workflow is clearly described.

The premise of this manuscript is simple: there are currently severe limitations on structural data as to how transcription factor (TF) bind their cognate DNA or nucleosomal templates in living cells. For example, ChIPseq reveals sequences that a given TF is associated with, but cannot distinguish whether the TF is binding directly to the DNA or via a complex of other DNA-binding TFs. In vitro experiments that involve recombinant proteins do not necessarily reflect in vivo results, and co-crystallization/NMR of protein-DNA interactions are not always possible, and when possible, may not reflect the native state of modified proteins/DNA/nucleosomes which bind a possibly modified complex of TFs.

To overcome this technical barrier to a complete understanding of how TFs behave, the authors have developed a novel approach based on xlinking mass spec. The method utilizes a high intensity UV femtosecond laser that can cross-link protein-DNA interactions, followed by digestion with nucleases and peptidases that cleave DNA/protein, followed by mass spec to discover the direct peptide-DNA interactions. This allows the interacting domains to be mapped directly, with higher resolution and accuracy, than more traditional means. To confirm, they used NF1 as an example and confirmed with this approach, and discovered interactions consistent with published biochemical literature. They also mapped additional sequences to the cytosines on the reverse TTGGA consensus sequence, a novel discovery not previously reported.

Overall, we thought this method/approach is novel and exciting, and will likely have widespread impact on research that relies heavily on crystallographic data that is incomplete/non-existent. That said, the work is really a Methods manuscript than a research/discovery article and should be treated as such. Otherwise, authors should elaborate on how the additional novel interactions impacts transcription and for which genes, which would require additional experiments such as ChIPseq and qRT-PCR to verify that their findings influence transcription in vivo (which we imagine they would prefer to avoid). If rewritten as a novel methods paper. this manuscript will be of interest to a wide readership and is suitable for the journal.

We thank reviewers #2 for their positive evaluation and agree that the manuscript has primarily a methodological focus.

We have two suggestions for improvement before acceptance:

1) The authors state they used in vitro reconstituted nucleosomes, but figures and methods describing how this was done are missing- for this kind of novel methodology, please provide these. This suggestion does not impact the central findings of the paper, but will increase

confidence in the interactome deduced from the ms/ms data.

For the reconstitution of recombinant nucleosomes, we followed the classic protocol of Luger (Luger et al. 1999, Chromatin protocols) with modifications of Dyer and Bartke (Dyer et al. 2004, Methods Enzymol, Bartke et al. 2010, Cell). To add more information on this part, we have included a sketch of the protocol together with quality control blots (SDS-PAGE and DNA retardation gel) in Supplemental Figure S2.

2) The authors used TiO₂ for the enrichment of the crosslinked peptides and subsequently eliminated all the other remaining peptides in their data analysis step. However, TiO₂ enriches many of the other types of modified peptides (phosphopeptides, ADP-ribosylated peptides) as well as acidic peptides, which may be ignored during the data analysis, but they are present in the spectra, therefore possibly suppressing the signal of the crosslinked peptides or skewing the quantification. Please comment on this point.

We purified our “in-vivo” samples in multiple steps that avoided the enrichment of phosphorylated or ADP-ribosylated peptides. These involved (i) biotinylation of digested chromatin, (ii) boiling under denaturing conditions, (iii) enrichment and purification of biotinylated DNA under denaturing conditions, (iv) elution of DNA crosslinked proteins by DNA digestion and (v) protein digestion and (vi) TiO₂ enrichment of crosslinked peptides. While phosphorylated and ADP-ribosylated peptides of non-DNA-crosslinked proteins are removed in step (iii), only phosphorylated or ADP-ribosylated peptides of DNA crosslinked peptides would be theoretically enriched. In fact, performing an open search in pFind (the search engine which to our experience reaches the highest depth of identifying post translationally modified peptides), indicates that only 3.4% of all enriched unique peptides are phosphorylated on serine or aspartic acid and no ADP-ribosylated peptide. All phosphorylated peptides were derived from three proteins - Hsp90b, Dcaf8 and Pairb - which can all bind directly to chromatin.

-This report was written by Drs. Yamini Dalal, Aleksandra Nita-Lazar and Minh Bui at the National Institutes of Health in Bethesda.

Reviewer #3 (Remarks to the Author):

This is a paper that builds upon a body of prior literature, dating back around 10-20 years by some of the authors, which explored the use of femtosecond laser irradiation for DNA protein crosslinking. In the description of the laser-based method, the paper is reminiscent of a recent study from the Altucci group ("Time resolved analysis of DNA-protein interactions in living cells by UV laser pulses", Scientific Reports, 7, 2017), who also explored the use of femtosecond lasers for DNA-protein crosslinking in situ (including transcription factors and histones). They didn't use mass spec, which is essentially the major innovation of the current paper. Normally

this wouldn’t be enough in the way of novelty, but “solving” the detectability problem for protein-DNA linkages has been a challenge, and so worthy of consideration if the solution is a durable one. There are some technical matters that dampen my enthusiasm, however. It is hard to get a good sense of how useful a development has been presented.

I have the following concerns:

1. The authors should be using scrambled oligos as a control wherever possible. As it stands, all we have to convince us that the interactions are legitimate is agreement with other studies. This is useful, but not a replacement for controls. In this context, dismissing the unexpected crosslinks (based on distance from the canonical binding sites) as being due to conformational change is an unsupported claim. It can also be due to diffusion of the activated species to sites of high reactivity, and thus not reflective the ensemble of conformers at all. A lot can happen during ~50 seconds of irradiation!

We agree with reviewer #3 and have now included controls with mutated consensus sites for NF1 (Figure 2d) and TBP (Figure 2f). As crosslinking formation to the mutated oligos was highly ineffective, we believe that our assay captures protein complexes in their native conformation.

2. There are other claims with weaker support than I think is justified, in the form of the identification of binding sites on the oligos. For example, what is the evidence that the cytosine crosslinks are specific for the reverse strand in the NF1 experiment (there are C’s elsewhere) .

We agree that the DNA crosslinked peptide (DCP) 14, which has a single cytidine crosslink, would not provide enough sequence information, to prove the binding specificity on the reverse strand. The annotated interactions in Figure 4C are derived from DCP13 (CCT crosslink) and DCP16 (AC crosslink), which only exist on the reverse strand of the consensus site.

Another unsupported claim is discounting Mad2I2 as the binding site for peptide KPLLEK, simply because it doesn’t fit with expectations for DNA binding.

In order to better identify the protein from which the KPLLEK peptide is derived, we have queried a newly measured, deep proteomic dataset of >9700 proteins and only identified Oct1 and Oct11 from all possible hits. We therefore believe that we can exclude the possibility that the crosslink would be derived from unspecific DNA binding of Mad2I2, as it is either not expressed or expressed at undetectable levels in our cell line.

3. What are the yields of crosslinked products? It would be useful to know how robust of a method this is from that standpoint, as the numbers of detected crosslinks are not particularly high. Is this because of LC-MS issues or the enrichment methodology?

We have included a quantification of the crosslink yield based on the Western blot of NF1. Under the applied conditions, about 10-15% of NF1 was crosslinked to DNA. We agree that for in-vivo fliX-MS the cross-link efficiency might not have been optimal and we are currently

working on improving it. In the current work, we irradiated cells in suspension. While the top layer of cells received full energy, the energy likely decreased towards the bottom of the tube, such that only a subfraction of cells were maximally irradiated. One possible solution is to use a laser with higher repetition rate in combination with a moving laser beam that allows to crosslink cells directly on the surface of a tissue culture dish.

4. Have all the modification types been considered? The software used was primarily designed for RNA-protein interactions.

We have modified the RNP-XL search to be specific for detection of crosslinks to DNA mono- or short oligonucleotides and derivations of such. We have expanded the Material and Method’s on that part.

5. Page 8, line 159. You cannot avoid DNA damage at these fluences. Minimize perhaps, but not avoid. You will also get protein-protein crosslinking between aromatic amino acids. These issues should be clarified.

We agree with reviewer #3 and rephrased the paragraph. We also mentioned protein-protein crosslinking throughout the result and discussion sections.

6. The identification of crosslinked sites is carefully done and reflective of the quality of the software used. There are a couple of puzzles though.

Line 244: is H₂O₂ loss of two OH’s? Or one water and an H abstraction?

The neutral loss of hydrogen peroxide on cysteine-containing peptides has not been reported yet. However, as we observed a full γ -ion series covering four amino acids including the cysteine that showed the H₂O₂ loss, we believe that there must be a chemical explanation to it. To us the most likely explanation would be the formation of an ethial-S-oxide (see below) although we cannot exclude that other chemical mechanisms take place.

Line 292: the evidence that both L1789 and K181 are crosslinked at the same time to the same cytosine is unclear. Could these not be chimeric spectra? It would be useful to convey the peptide retention times for this (and the other features). Based on the chemical scheme presented in figure 5g, I find double coupling unconvincing.

We revisited the data and identified an alternative explanation for the doubly crosslinked peptide. The most intense peak in the low m/z range ($m/z = 89.06$) corresponds to deoxyribose after loss of CO, which can be seen as a fragment ion of the deoxycytidine -CO ion, after the loss of cytosine. This ion provides evidence that the CO adduct on the product ions results from a crosslink to the deoxyribose part of the nucleotide, rather than being derived from cytosine fragmentation. Therefore, our data suggest that one crosslink targets the ribose part of the deoxycytidine and the other crosslink targets the nucleobase, and we have changed the proposed model in Figure 5g accordingly.

Following the suggestion of reviewer #3, we have investigated the possibility of a chimeric spectra. For the given retention time the isolation windows is depicted below:

There is no other precursor available in the isolation window that would be proportional to the ratio of specific fragment ion peaks for both adducts in the MS2 spectrum. Therefore, we can exclude the possibility that this data represents a chimeric spectra.

7. The discussion section adds very little to the paper, as it is really a restating of the results section, and the claims in the final paragraph seem a bit overstated.

We have expanded the discussion section and elaborated on possible limitations of our method. We agree that the outlook was a bit overstated and changed the word ‘entire’ to ‘global’.

Editor (additional remark)

In addition to addressing all of the reviewer requests, we also ask you to describe in more detail whether/how FDR control was performed for the crosslinks identified with RNP(xl).

We performed alignment of control and UV-irradiated sample and filtering of the UV-irradiated sample as described (Kramer et al. 2014, Nat Methods). Briefly precursor ions in the UV-crosslinked were removed if they had less than 2-fold higher intensity than in the control sample within a retention time window of 0.33 min for TF-DNA complexes and in-vivo fliX-MS and 0.65 min for nucleosomes. Next, spectra that could be assigned to a non-crosslinked peptide were filtered out with a false-discovery rate of 1% (peptide ID filtering). Although we found RNP-XL to be an efficient tool for peptide selection, the list of proposed crosslinks needed to be carefully curated manually, based on the criteria mentioned in the Material and Method section. We envision that a future software that enables automatic detection and filtering of DNA-crosslinked peptides - based on the criteria and mass shifts described in this study - would have a great advantage to the field and will enable fliX-MS analyses on global level.

Reviewer #1 (Remarks to the Author):

Overview:

More than thirty years ago pioneering studies by the J.T. Lis laboratory in *E. coli* and *Drosophila* (Gilmour and Lis, 1984, PNAS 81; Gilmour, D.S. et al, 1986, Cell 44) demonstrated the potential power of UV irradiation mediated protein DNA crosslinking for studying the biology of DNA binding proteins. As a matter of fact these papers are also the first ones describing a kind of chromatin IP assay, which is nowadays in most of the cases carried out employing formaldehyde (FA) crosslinking. Follow-up work by the laboratories of M.D. Biggin, D. Houde and G.L. Hager (Walter, J. and Biggins, M.D., 1997, Methods 11; Moss, T. et al., 1997, Methods 11; Nagaich, A.K. et al., 2004, Mol. Cell 14) demonstrated that the efficiency of UV DNA protein crosslinking can be increased by utilizing pulsed UV light emitted from lasers. Concomitantly, literature reports from the last 20 years also suggest that the technological evolution of laser technology (e.g. the move from nanosecond to picosecond laser pulsing technology) bear the potential to address the major shortcoming of UV mediated DNA protein crosslinking, namely its low efficiency that comes along with extremely low recovery of covalently bound DNA protein complexes hampering the general applicability of the approach.

This is in stark contrast to the field of RNA protein interactions for which UV crosslinking has been and still is the mainstay of any genomics or proteomics follow-up methodology. Until now, the huge difference in the yield of UV-crosslinked RNA-protein versus DNA-protein complexes is explained by the compromised photo activation of the nucleobases in the double stranded DNA (base pairing, base stacking) versus the situation found in single stranded RNA. Counteracting this by prolonging the exposure time (using standard UV lamps) is not feasible because under such conditions severe photo damage of biological macromolecules prohibits subsequent biochemical workflows (e.g. cells cannot be efficiently lysed anymore).

Therefore, FA crosslinking, which exhibits recovery of DNA-bound proteins in the range between 0.01-2.0% and that forms metastable zero-length covalent DNA-protein complexes (but is partially also recovering piggy-back/indirect DNA-protein interactions) is currently the commonly employed starting point of any large scale experiments like ChIP-seq, FAIRE-seq and Hi-C/3C-seq.

In this context the proposed manuscript by Reim and colleagues tries to overcome the current bottlenecks of examining DNA-protein interactions with the help of UV irradiation by introducing the application of femtolaser pulse technology in conjunction with state-of-the-art nanoLC-MS analysis of DNA-crosslinked tryptic peptides. At first the authors claim extremely high crosslinking efficiency (up to 100%) in reconstituted *in vitro* systems but the efficiencies of crosslinking in live cells as well as quantifying the ability to recover the UV XLinked molecules are not addressed.

Nevertheless, this is with no doubt a very interesting and pioneering study which should be published in a decent journal like Nature Communications. Emphasizing the challenging task of mapping the interaction surface between proteins and DNA, this is much needed for the field. The used approach is innovative, appropriate and a step in the right direction but for reasons outlined below a couple of issues mainly related to (quality) control experiments, data representation, the

need for a much better and detailed description of the methods and a more critical discussion have to be resolved in a major revision prior to publication of the work.

In addition and since this is important for the editor to judge future citation of this study, the general availability and robustness (easy to replicate?) of the experimental femtolaser setup is currently not clear at all. In other words, could Wierer, Reim and coworkers envision that their technology will become available on a routine basis in the near future?

Therefore, I would like to recommend the work of Reim and colleagues for publication in Nature Communications given that the important concerns rose above and detailed below can be rectified in form of a major revision.

We are pleased by the positive and constructive feedback of reviewer #1 and especially thank him or her for placing our work in a historical context.

Thank you very much! Overall and as outlined above and below, the authors have addressed most of the reviewer's questions and suggestions in the revised manuscript.

Major points:

Description of methods: the materials and methods section is lacking information that is crucial for both understanding the paper and the possibility to reproduce the work.

☐ In case of the synthetic oligonucleotides it is not clear if the 5'-biotinylation is only present at the "sense" strand or on both (also reverse) oligos?

The 5'-biotinylation is present on both forward and reverse oligo. We have included this information in the new material and methods section.

Great!

☐ In general, it is difficult to extract information concerning the concentration and total amount of protein(s) and DNA in the reconstituted in vitro systems (NFI, TBP, reconstituted nucleosome core particles)

We have expanded the information regarding the exact amounts and concentrations in the material and methods section.

Perfect!

☒ The LC/MS methods do not provide the following parameter settings: source temperature, RF-value for S lense (or equivalent of it), target value and injection time for MS1

We have improved the description of LC/MS methods including the missing information

Very good!

☒ The description of the data analysis is lacking the following parameter description: number of max. allowed missed cleavages, PD and RNPxl version and the criteria applied for manual curation (e.g. what is the percentage of spectra initially proposed by PD/RNPxl that had been manually removed?)

We now include more detailed information on maximally allowed missed cleavages, version information for Proteome Discoverer and RNPxl as well as a detailed description of the criteria for manual spectra curation in the material and methods section. Due to the very stringent filter criteria the number of spectra that remain after filtering the list of initially proposed cross-links by RNP-XL is < 1% of the original list. Therefore, we do not use RNPxl as full analysis tool, but rather as an initial filter for spectra identification. We believe that a future software, which can automatically detect DNA-crosslinked peptides based on the criteria described in our manuscript, would be a highly valuable tool, and we are currently envisioning such a development as an add on to the new AlphaPept software that will be published in a few months.

Well done!

☒ The sketch of the femtolaser setup (see Fig. 1) and its description makes it complicated to compare the work presented by Reim et al. with the literature (picolaser or nanolaser configurations etc.). Please indicate the pulse repetition rate (is it 0.5 MHz?), the origin/manufacturer of the b-barium borate crystal and the configuration of the active fiber systems laser (“high-repetition rate” or “high pulse energy”). It would be very beneficial for the general reader if the authors could provide in the

main text an idea on how much average power (Watt) and peak power (Watt) is actually applied to the samples (also W/cm²). Moreover, it would be nice to clarify (for their standard parameters of 500 fs pulses at 50 nJ and a total energy of 1.25 J) how many pulses (equal to the total exposure time of the biological samples) are actually required (about 5,000? ~10 ms exposure time?). I am asking this because until now it seems that there is no standard in presenting such data and older literature is often using W and W/cm² in the figures.

We have greatly extended the description of the laser setup in the material and methods section including all variables requested by reviewer #1 and updated figure 1 as suggested.

I would like to thank the authors for this detailed explanation and appreciate the changes in the manuscript.

Finally, estimation on how many photons per nucleotide are actually needed to reach saturating photo crosslinking of NFI to the synthetic DNA would be quite nice to enhance the understanding of the work by non-physicists.

Quantum efficiencies in general and the number of photons per nucleotide in our case are very helpful parameters to quantify classical photochemical processes based on the absorption of single photons. Several particularities complicate the classical photochemical description of the laser-induced processes, which are described in the publication. These are:

Femtochemistry: Due to the high intensities used in the femtosecond range and the underlying 2-photon processes, non-linear intensity dependence of the absorption processes occur. Moreover, the absorption cross section is still unknown.

Biological model: The paper uses 'xlinked species' as parameter to quantify xlink yield (Figure 2). Xlinked species describes the covalent binding of the transcription factor to the DNA and can be achieved by at least one xlink of a nucleotide to the amino acids in a DNA-Binding sequence. We have identified five crosslinks locations on the palindromic consensus DNA-binding sequence of NF1 exist (see Figure 4c), which can potentially contribute alone or together with others to the parameter 'xlinked species'.

Therefore, we believe that due to the complex, underlying physical processes with non-linear dependencies and the chosen method for determining the cross-linking efficiency, an estimation of the quantum efficiency is afflicted with too high inaccuracies and requires too many assumptions.

Thank you very much for clarifying this complicated matter. Hence, I agree with the conclusion of Reim et al. Maybe it escaped my notice but I cannot find a short but comprehensive description of

the concept 'xlinked species' in the revised manuscript? It would be nice to add this to the final version!

Data availability: the data are uploaded to the EBI PRIDE data base but unfortunately no reviewer login credentials have been communicated. Please allow the reviewers to access the .RAW files during the revision process. Additionally, it would be helpful to provide the following detailed information: for each of the annotated MS/MS spectra presented in the main figures and figure S2 please indicate the name of the corresponding .RAW file and the scan number so that reinterpretation will be straightforward.

This was provided already but we apologized if it was not in an obvious place: The PRIDE login information is as follows: Project accession: PXD014898, Username: reviewer75731@ebi.ac.uk, Password: VVhUy6lh

We have now included the raw file name and scan number relationships of all presented spectra in Supplemental table 1.

I apologize for having missed this information. Thanks a lot for providing the raw file name and scan number relationships. This eases the process of manual inspection.

Figure 1:

☐ Figure 1a: please consider the comments above and enhance the sketch so that the general reader is immediately able to perceive the main advantages of the femtolaser system for biological systems.

We have updated Figure 1 to make it more conceivable to the general reader.

Thank you. Please reconsider "Fig. 1C: RNPXL and manual annotation" The MS spectrum is not really readable (in a print out). Maybe a cartoon (hypothetical spectrum) emphasizing the detection of peptide ion fragments and peptide-nucleobase fragments as well as nucleobase ions would do a better job?

☐ Figure 1b: It might be helpful to mention the DNA digesting enzymes in the legend.

We have included the information on digesting enzymes in the Figure legend.

Great!

Besides, I have some minor concern regarding the combination of enzymes utilized. Benzonase is leading to digestion products ranging from mononucleotides (rare) to short DNA oligonucleotides (2 to 5 nucleotides) that all carry a 5'-monophosphate. Likewise, DNase I produces oligonucleotides harboring a 5'-monophosphate. In contrast, MNase results in the generation of mononucleotides and short oligos possessing a 3'-phosphate. It is known for some RNases (but to my knowledge not for Benzonase and MNase) that they can also work as (ribo)nucleotide phosphatases. Hence the combination of Benzonase and MNase might reduce (in case they exhibit phosphatase activity) the amount of phosphorylated mononucleoside-peptide heteroconjugates thereby prohibiting their interaction with TiO. Do the authors have some indication that mononucleotide crosslinks are underrepresented in their data set? The use of C18 STAGE tips might lead to the loss of some peptide-DNA heteroconjugates (if the overall hydrophilicity is high). Please discuss this possibility (e.g. an additional thin layer of Perseptive OligoR3 material on top of the C18 sandwich might lead to an improvement). Importantly, the authors completely skip to consider that in case of "in vivo" samples the TiO beads will also retain other posttranslationally modified peptides like phosphopeptides. Did the authors assess in their data set obtained from the murine ES cell chromatin, if their workflow is also enriching phosphorylated or O-glycosylated peptides? Please discuss this.

These are very interesting notes. We have intensively queried pubmed and other sources to find out whether MNase or Benzonase might possess phosphatase activity, and did not find a published evidence for this. Although we cannot exclude it completely, we believe that there is not such activity. About 35% of all crosslinks identified in this study are mononucleotides, suggesting that our method is capable of capturing such species.

The use of OligoR3 is an interesting proposal, which we will investigate in future. As RNA-crosslinks and hydrophilic phosphopeptides are commonly purified by C18 material (for instance see Kramer et al. 2014, Nature Methods), we believe that this purification method would be well suited for DNA as well, although we are very motivated in testing alternatives. In addition to OligoR3 we could also imagine graphite columns as an alternative, which are known to favor hydrophilic peptides over non-hydrophilic ones (Larsen et al. 2004, Mol Cell Proteomics).

We purified our “in-vivo” samples in multiple steps that excluded the enrichment of phosphorylated peptides. These involved (i) biotinylation of digested chromatin, (ii) boiling under denaturing conditions, (iii) enrichment and purification of biotinylated DNA under denaturing conditions, (iv) elution of DNA crosslinked proteins by DNA digestion and (v) protein digestion and (vi) TiO₂ enrichment of crosslinked peptides. While phosphorylated peptides of non-DNA-crosslinked proteins are removed in step (iii), only phosphorylated peptides of DNA crosslinked peptides would theoretically be enriched. In fact, performing an open search in pFind (the search engine which to our experience reaches the highest depth of identifying post translationally modified peptides), indicates that only 3.4% of all enriched unique peptides are phosphorylated on serine or aspartic acid and no other phosphorylation type. All phosphorylated peptides were derived from three proteins - Hsp90b, Dcaf8 and Pairb - which can all bind directly to chromatin.

I am fully content with the elaborations of the authors.

Figure 2:

☒ Figure 2a: Maybe label Fig. 2a with “EMSA”. Which conditions (ratio between protein and DNA) were chosen for the experiments depicted in 2b-d? The EMSA lacks a specificity control in form of a double-stranded DNA oligo harboring a mutated NFI binding site. Alternatively, the binding reactions could be competed by an increasing amount of non-biotinylated DNA bearing a mutant NFI binding sequence.

We have marked the blot as “EMSA” and included detailed information on amount and molar ratios in the material and methods sections.

We have also repeated the experiment including DNA comparing wild type and mutated NFI binding site. Please note that NF1 also has unspecific binding (Dekker et al. 1996, MCB), which likely explains the faint shifted band for the mutated DNA oligo.

Well done!

☒ Figure 2b: Maybe label all immunoblots as “Western Blot”.

We followed the recommendation for labeling the blots as suggested.

Perfection!

Please provide the amount and concentration of DNA and protein.

We now include this information in the material and methods section.

Perfection!

The authors claim that this figure unequivocally demonstrates the formation of UV crosslinked NFI DNA species that seem to migrate in two major forms in the denaturing SDS-PAGE. Since the read out of the gel is detecting the biotinylated DNA this is an over-interpretation of the data as long as DNA-DNA photo-crosslinking cannot be ruled out. This experiment has to be repeated with DNA bearing the WT NFI site and control DNA harboring a mutant NFI binding sequence. In addition, optional proteinase K treatment of some of the binding reactions/samples will clarify if the visualized gel bands change mobility in a protein-dependent manner. In this context the authors should also consider that the molecular weight of the putative crosslinked DNA-protein species differs in Fig. 2E (clear discrepancy!). In the latter case the read out of the gel bands is protein-centric (anti-His-tag antibody reaction with His-tagged NFI).

We appreciate the comments and added additional controls for this part. First, we repeated the total energy curve experiment using DNA oligos harboring either a NF1 consensus sequence, or a mutated form of it (Figure 2d). This showed that the signal in the DNA-centric Western-blots is clearly dependent on specific interaction of NF1 with DNA. Second, to clarify the discrepancies in molecular weight, we performed a Western Blot for the UV crosslinked NF1-DNA complex after digest with or without prior proteinase K or DNase I treatment (Figs. 2e, S1d). After detection of the His-tagged NF1, we stripped the membrane and probed it with the Streptavidin-HRP conjugate to detect biotin-labeled DNA. We realized that about 7.5% of NF1 crosslinked to single- or double stranded DNA under these conditions, while a major part (53%) shifted to higher molecular weight level, indicating that protein-protein crosslinking does take place (Fig. S1). In fact, the high molecular weight band at ~115kDa corresponds to two NF1 molecules, while the two lower bands fit to the molecular weight of NF1 bound to a single or double-stranded NF1 oligo. Notably, the signal of the high-molecular weight fraction disappeared with DNase I digestion, indicating that the signal represents protein-DNA crosslinks. While the mono-NF1-DNA crosslinks disappeared with proteinase

K treatment, the crosslinked bands in the high molecular weight region are not well enough resolved, to allow differentiation between protein-protein and protein-protein-DNA crosslinks. Extrapolating from the crosslinking efficiency of mono-NF1-DNA and the intensities of the 65 of kDa, 115kDa and 185kDa bands in the DNA-biotin blot, we estimate a crosslinking efficiency of 14% under the given energy conditions.

To improve readability, we decided to show only the bands corresponding to the mono NF1-DNA crosslinked complex throughout Fig. 2, and provide the full blots as supplemental data in Supplemental Figure S1.

Very nice. These additional experiments now fully explain the results!

Finally, the choice of total energies is not ideal for curve fitting and more measurement points should be provided, especially in light of the fact that the authors are claiming a two-photon process.

Our new energy curve (Figure 2d) includes more data points and shows that saturation is reached already at about 316 mJ. With increasing total energy the generation of higher molecular weight species predominates leading to a decrease in intensity of the mono NF1-DNA band.

Thanks for making these data more robust!

☒ Figure 2c: same comments and concerns as for Fig 2b.

Please see the answers just above.

Agreed!

☒ Figure 2d: Again, a series of control samples employing DNA containing a mutant NFI site are crucial in order to assess if the femtolasers crosslinking reaction can discriminate between sequence-specific and general DNA binding. Otherwise, the application of fliX-MS to biological chromatin samples would face the problem of recovering functional and non-functional protein DNA interactions.

Please see answers above just above.

Agreed on!

Therefore, Reim et al. should also discuss the potential utility of their method for interrogating DNA protein binding reactions conducted in the presence of cellular or nuclear extracts instead of using highly purified recombinant protein.

Although we thought about the possibility to study protein-DNA binding with nuclear extracts, we decided that UV-crosslinking of cells would be a better way to show that the method is capable to capture more 'in vivo' protein-DNA interaction events, given its fully unbiased character. Nevertheless, we believe that the proposed experiment should be feasible for clarifying DNA interactions for proteins that have a known and specific consensus site, but are difficult to purify and we now include the potential utility of this in the discussion section.

Thanks for considering this!

As proposed for Fig 2b nuclease digested samples would provide additional evidence that the visualized gel bands are indeed covalently attached DNA-protein complexes. This would help to argue against certain literature reports (for example Itri, F. et al., 2016, Cell. Mol. Life Sci. 73, 637-648) that claim protein-protein crosslinking induced by UV laser irradiation.

Please see answers above.

Fine!

In summary, without the proposed improvement of the data in Fig. 2 the conclusions drawn on the crosslinking efficiencies would have to be considered preliminary. Hence, it will be absolutely essential to improve the manuscript for this paragraph in order to be able to better judge the future impact and possible shortcomings of fliX-MS.

Integrating the suggestions of reviewers #1 and #3 we believe that this part of our paper strongly improved from them.

I concur with the statement of Reim and colleagues and would like to conclude that they have sufficiently addressed the concerns and comments of reviewer #1 and #3!

Figure 3:

☐ Figure 3a: Please refer to comments stated below (concerning data presentation in the supplementary material). Please add an additional column, which provides information on the charge state, m/z value and the “mass accuracy” (mass deviation in ppm) of the parent ion.

We have included information on charge state, m/z value and mass accuracy deviation in the table presented in Fig. 3a.

Very good!

☐ Figure 3a, b: and in general: Albeit the fact that the total number of identified crosslinked DNA-peptide spectra is limited, it would be nice if the authors could mention if they were able to spot preferential crosslinking of certain amino acid side chains?

We analyzed preferential crosslinking among 14 DNA-crosslink peptides, where the crosslink unequivocally mapped to a single amino acid. While we identified an underrepresentation of amino acids with aromatic side chains (top panel), this effect disappeared, when normalizing on the amino acid frequencies reported by the most recent Uniprot release (2019_11) (bottom panel). Most notably, we observed a complete absence of DNA crosslinks on acidic side chains, however, given the overall low statistical power of this analysis, we decided to not include it in the manuscript.

Thanks a lot for this nice piece of information. Under these circumstances I follow the suggestion of the authors to not include this as a figure.

☐ Figure 3c: I highly recommend discussing the expected but missed-out histone DNA interactions.

One limitation of our current assay format is the use of trypsin for protein digestion, restricting the potential detection to tryptic peptides. Due to the high percentage of lysine and arginine residues, histones are particularly challenging for proteomic analyses. Together with effects derived from different crosslink efficiencies, this contributes to the non-complete coverage of histone DNA interactions. We included an explanation and possible solutions in the discussion section.

I agree with the authors that these technicalities (absence of a chemical propionylation step prior to trypsin digestion) could provide a good explanation for the missed out crosslinks. But other scenarios might also come into play here. Some of these tryptic peptides are indeed too short in order to be captured by C18 in general. Others (peptides with a length of five to six amino acids) are often excluded by standard proteomic data processing approaches or are not retained by C18 STAGE tips (short and very hydrophilic ones). Therefore, I would like to recommend that the authors have a look into their data with the goal to determine which of the non-crosslinked peptides are already missing (escaping their detection).

Overall a supplementary figure showing some standard assay(s) characterizing the quality of the in vitro assembled recombinant histone octamers would be helpful.

We have included a Coomassie stained SDS-PAGE and DNA retardation gel comparing free and nucleosome bound DNA in Figure S2.

Great!

Figure 4:

☐ Figure 4a: Again, the charge state, the m/z value and the “mass accuracy” should be shown.

We have included the missing information in the table presented in Fig. 4a.

Perfection!

☐ Figure 4d and e: Normally one would assume that the formation of a covalent bond between an arginine side chain (or lysine side chain) and a bulky molecule like di- or tri-nucleotides would prevent the cleavage of this site by trypsin or LysC. Therefore I would suggest reconsidering the interpretation of the spectra. The $\gamma_{1+C'-A}$ ion in Fig. 4d is very weak and normally one would expect a strong γ_4 ion because of the preferential gas phase cleavage N-terminal to prolines. Maybe the crosslink is at a different position in the peptide, e.g. the internal arginine? Besides, the “ γ_7 ” is the singly charged (mono-protonated) peptide precursor ion that had lost DNA. The same holds true for “ γ_6 ” in Fig. 4e. There is also no need to consider that the crosslink is present at the C-terminal arginine residue in the spectrum in Fig. 4e.

We agree with reviewer #1 that a crosslink on the terminal arginine would actually interfere with trypsin digestion. In fact, as the intensity of the γ_{1C} ion is very low, and as it represented the only evidence for a crosslink location on R89, we removed the localization annotation for this peptide. Likewise, we agree that the crosslink localization on the terminal arginine in DCP15 (Fig. 4e) is unlikely as well. As in this case the crosslink information is based on the presence of an arginine immonium ion, we were able to locate the crosslink on R117 instead. We have updated Fig. 4 and the text accordingly.

Yes, this interpretation of the spectra makes much more sense!

☐ Figure 4f: Could the authors please specify the molecular identity (formula) of the trioxidized cysteine? I assume that they are referring to cysteic acid that is generated via sulfenic and sulfinic acid intermediates.

The trioxidation we refer to is indeed cysteic acid (Williams et al. 2011, J Am Soc Mass Spectrom). We clarified this in the text.

Thanks a lot!

Is this modification already present in the recombinant protein before UV irradiation or do Reim et al. have evidence that this is a photochemical process? In case of the latter please be reminded that photo oxidation of adenosine (oxidized A) would be isobaric to G. Hence the interpretation of the crosslinked nucleotide mass shift could potentially be attributed to AoxT-HPO₃.

Although we cannot exclude that oxidation might occur during UV-radiation, we believe that cysteine trioxidation occurs during sample preparation similar to oxidation of methionine. Standard proteomic workflows include reduction and alkylation of cysteine residues (which would remove the modification). Here, we chose not to include these steps to avoid losing potential crosslinks that would be reduced by DTT or introducing alkylation artifacts. In fact, trioxidation of cysteine has been reported as a common side reaction in standard buffer conditions during MS sample preparation in absence of reduction / alkylation (Bayer and König, 2016, Rapid Commun. Mass Spectrom.). In addition, revisiting our data we identified cysteic acid containing peptides also in the control samples that were not UV-irradiated. We therefore changed the text accordingly.

Many thanks for making this much more clear!

☐ Figure 4g: Again y12 is equivalent to the peptide molecular ion (in this spectrum the doubly charged species) that had lost C. The proposed C⁻ ion exhibits indeed strong ion abundance. Nevertheless, in the RNA field, nucleobase ions are rarely observed in case of mononucleotide crosslinks, because the nucleobase is believed to be covalently bonded to an amino acid side chain. Hence, observation of this ion will only be feasible in case the bond could dissociate during HCD (e.g. an amide bond crosslink between the amino group of C and the aspartate side chain).

While the free nucleobase could indicate a crosslink via the deoxyribose part, the existence of a nucleobase crosslink on the y3 ion clearly shows that the crosslink is nucleobase directed. Hence, we believe that the free nucleobase must indeed be dissociated from the crosslinked peptide during HCD fragmentation.

Yes, I would also BELIEVE so. Nevertheless, it should be made clear that this is a hypothesis.

With the goal to ease data interpretation of expert readers the authors should provide (supplementary information) spectrum ion tables (as generated by DB search engines like Mascot, Protein Prospector or PEAKS studio) containing the theoretical fragment ions ((a-), b-, y- and nucleobase shifted series; major expected neutral loss peaks) and highlighting the assigned peaks. The same applies to Figures 5, 6 and Supplementary Figure S2!

This is a very good point and we have now included detailed tables with theoretical and observed fragment ions along with observed mass shifts and mass accuracies for all reported MS2 spectra as Supplemental file 1.

Well done! Very informative!

Figure 5:

☒ Figure 5a: Same comment as for Fig. 4a.

We have included information on charge state, m/z value and mass accuracy in the table presented in Fig. 5a.

Nice!

☒ Figure 5b: Fig. 5a lists three putative crosslinks at single amino acid resolution and two crosslinks at lower resolution but the chart just displays four of them.

This discrepancy is due to the proposed double crosslink of the LDLKTIALR peptide to cytosine.

OK!

☒ Fig. 5d: Honestly, I am unable to find any possibility for UV crosslinking the dinucleotide AC (or CA) in the sequence window depicted in Fig. 5b. There is only one AC (and one CA) a bit more

downstream of the TATA box and one CA upstream of the TATA box (not shown in Fig. 5B; all present on the antisense strand) in the short synthetic oligo derived from the AdML promoter. Alternatively, Reim and colleagues try to propose the presence of two separately crosslinked mononucleotides {[A]+[C]}-HPO₃, namely A (from the sense strand) and C (antisense/reverse strand)? This discrepancy or confusion must be resolved during revision.

We agree with reviewer #1 that the annotation of this crosslinked peptide was not clear and have now improved it. Our data indicates two independent crosslinks to A and C on the same peptide, which are evidenced by 3 and 4 unique shifted product ions, respectively.

Thanks for improving this!

▣ Fig. 5e and g: First of all I would like to draw the attention to the low m/z range window of the MS/MS spectrum, which contains many strong peaks that are not assigned.

The peaks in the low m/z range are sub-peptide size and therefore likely fragmentation products, which are difficult to annotate. Individual analysis of these peaks identified one as the immonium ion of lysine, and another one as deoxyribose after loss of CO (see below).

OK!

Second, the generation of b₁ and therefore a₁ ions depends on the presence of “unusually” strong acylium ions that are - based on the classic work of Schlosser and Lehman - normally not observed in charge directed (mobile proton mediated) fragmentation pathways with the exception that a basic amino acid (K, R and H) is present at the peptide N-terminus (Hiserodt, R.D. et al., 2007, J. Am. Soc. Mass Spectrom. 18, 1414-1422) or the amino terminus is chemically modified (e.g. N-terminal acetylation). Probably, the authors are able to propose a hypothesis why the crosslinked C nucleobase could stabilize the b₁/a₁ ions.

This is a very interesting note. However, we believe that the discrepancy results from the use of HCD fragmentation in contrast to low-energy CID fragmentation that was applied in the above mentioned work. A more recent study from the Yates group (Diedrich et al. 2013, J Am Soc Mass Spectrom) studied the fragmentation series of peptides in response to increasing HCD energy observing a strong increase of smaller a and b ions with increased energy. In fact one of the peptides in the study

(LTILLEELR) contained an N-terminal leucine similar to DCP20. Starting at a normalized collision energy of 20, the authors observed a strong presence of a leucine immonium ion, which is indistinguishable from the a1 ion.

Another study from our group (Michalski et al. 2012, JPR) also systematically investigated HCD fragmentation spectra and reported an a1 ion representing a phosphorylated N-terminal serine. We therefore believe that the deoxycytidine crosslinked to the a1 ion should be equally possible to be generated by high-energy HCD fragmentation.

This is a reasonable explanation and warrants more detailed mechanistic studies on HCD fragmentation (gas phase chemistry). I have to admit that the Dietrich et al. paper has escaped my notice!

Third, even though I appreciate the clever solution for finding an interpretation of the MS2 spectrum in Fig. 5e and g, I would like to mention that up to my knowledge such a mechanism has never been proposed by the RNA field, or is this “plus CO” adduct an optional parameter in the newest version of RNPxl? I would also like to remind that mass spectrometry as such and without further help of chemistry experiments (e.g. chemical reactions carried out with the molecule of interest followed by MS analysis) is far from being a perfect technique for de-novo elucidation of chemical structures. Therefore, currently the data interpretation in Fig. 5e leads to a nice hypothesis that should be corroborated by repeating the experiment with a synthetic ds-DNA oligonucleotide harboring a [13C,15N]-labeled C at the suspected contact position (with TBP). The Munich area based company SILANTES should be able to provide this reagent. Alternatively, the utilization of recombinant SILAC-labeled (at arginines and lysines) TBP or tryptic digestion in the presence of [18O]-H2O would at least serve as a prove for the identity of the $\gamma_6+\text{CO}$ to $\gamma_8+\text{CO}$ series as well as the peptide ion+CO peak assignment.

We followed the suggestions of reviewer #1 and ordered an oligonucleotide, in which the first C on the complementary strand downstream of the TATAAAA sequence was isotopically labeled by 13C and 15N. We repeated the experiment using this oligo, however due to technical problems on the mass spectrometry side we were not able to analyze this experiment and we were not able to access the laser in the time-frame of the revision for a repetition of the experiment (also due to the initial long synthesis time of the oligo).

Especially, in the current situation I can fully understand that experiments cannot easily be conducted in a reasonable time frame anymore.

However, revisiting the data, especially the peaks of the lower mass range, yielded an interesting observation, which points to an alternative model for generation of the CO-crosslinked product ions.

The most intense peak in the low m/z range ($m/z = 89.06$) equals deoxyribose after loss of CO, which can be seen as a fragment ion of the deoxycytidine -CO ion, after the loss of cytosine. This ion provides evidence that the CO adduct on the product ions results from a fragmented deoxyribose, rather than being the product of cytosine fragmentation.

Taken together, our data indicates that one crosslink targets the ribose part of the deoxycytidine while the other crosslink targets the nucleobase. We therefore changed the proposed model in Figure 5g accordingly and updated the text.

I am able to recapitulate the calculation of the authors (CO: 27.99491463 amu) resulting in a value of m/z 89.06026 for the proposed ion species. Radical based (induced by UV/oxygen) activation of the deoxyribose C5 position has been described in the context of DNA damages. In addition, the loss of cytosine or nucleobases in general (dissociation of the N-glycosidic bond) are well documented as well. But the molecular pathway resulting in the formal adduct formation equal to the monoisotopic mass of CO remains enigmatic. In consequence, Reim et al. are again just proposing another interesting hypothesis. Moreover, the authors readily admit that there exists a large distance between L178/K181 and the C in the crystal structure. The latter might indicate structural flexibility. Nevertheless and in light of the fact that the manuscript should mainly provide proof-of-principle experimental results, I strongly recommend to move Fig. 5e, f and g to the supplementary section. Likewise, it would make sense to remove the L178/K181 crosslink from the current figure 5f. In my point of view these results are not essential for convincing the general reader about the novelty and technological relevance of the study!

Last, I would like to recommend changing the wording in the main text (line 286 to 288). Of course, the CO adduct is not present in the parent ion containing the C-HPO3 crosslinked desoxyribonucleotide (heteroconjugate of LDLKTIALR and C-HPO3). Therefore, the sentence is rather confusing.

We agree that the sentence was not very clear and removed it.

Thanks!

Figure 6:

☐ Figure 6d: the authors could conduct an in depth proteome analysis of the chromatin input as well as check public repositories for murine ES cells (e.g. RNA-seq) with the goal to find out if all the

possible proteins (Hes2, Oct1, Oct2, Oct11 and Mad2l2) are present in their chromatin sample or if they are even expressed.

We appreciate this suggestion and included a deep proteomic measurement (> 9700 proteins) of the same ES cell line (Supplemental Figure S4b) than the one used for fliX-MS. The exclusive presence Oct1 and Oct11 in this dataset suggests that the identified crosslink derives from either of these two transcription factors.

This is a very helpful additional experiment/dataset!

☐ Figure 6e: Maybe reference this paper (Winter, D. et al., 2010, J. Am. Soc. Mass Spectrom. 21, 1814-1820) to explain the presence of the c1 ion.

We followed the suggestion of reviewer #1 and included the reference in the manuscript.

Great!

Supplementary Figures:

☐ Supplementary Figure S2d: It might make more sense to assign the dinucleotide as [TA+HPO3] because the internal fragments are shifted by a thymine adduct and there is a strong peak for the nucleobase A`.

We have modified the figure accordingly.

Perfect!

☐ Supplementary Figure S3: there is one peptide ion (from HES-2, KPLLEL + [TT], MH33+) that exhibits a Δm of 10.7 ppm even though the methods section claims that the DB search was performed with a maximum allowed mass deviation of +/- 10 ppm.

This peptide was identified by RNP-XL as an AA-H₂O crosslink. However, as the spectra contained a marker ion for thymine, and the mass of the AA-H₂O adduct is very close to TT (626.1152 vs. 626.1027), we manually annotated the peptide as a TT crosslink, even if the mass deviation slightly exceeded the 10 ppm limit.

The mass deviation between the RNP-XL proposed AA-H₂O crosslink and a TT adduct is substantial (almost 20 ppm)! What was the (measured) mass deviation for the proposed AA-H₂O crosslinked species?

Generally speaking, it would be important to know if the MS₁ spectra were recalibrated at the global level (as performed for instance by MaxQuant)? Since on one hand the PD version is not indicated in the methods section and on the other hand I am not familiar with the details of PD data processing this is an important point because many spectra do actually show a Δm of greater 5 ppm, which at least in my experience is unusual for correct IDs on data acquired on a HF-type orbitrap instrument, except recalibration was not carried out. In this context I would also like the authors to provide the Δm values for the major fragment ions in the ion spectrum tables requested in my comments above (Figure 4). Information on the latter will provide additional trust in the manual curation of MS/MS spectra.

To our knowledge and based on the documentation, the RNP-XL workflow does not contain a recalibration step. We have followed the suggestions of reviewer #1 and included Δm values in Supplemental table 1 containing all theoretical and identified fragment ions.

This is very nice. In general, most of the fragment ions have a very low mass deviation. Still, quite some of the precursor/parent ion masses exhibit Δm values of greater 5 ppm. Apart from the absence of a recalibration step (in the workflow) this is maybe also a problem related to Proteome Discoverer software underperforming in accurate m/z determination from DNA-peptide heteroconjugates (contain a relative higher number of the 'mass deficient' oxygen atom as compared to amino acids/peptides)? In other words do the authors observe a better fit to the theoretical mass for "normal" tryptic peptides?

Minor points:

Title:

The utilized high resolution MS is nowadays standard and not worthwhile to be emphasized.

We have remove the phrase 'high-resolution' from the title.

Great!

Figure 1:

☐ Figure 1a: SHG (please explain the abbreviation in the legend)

We have replaced SHG (second harmonic generator) by 'borate crystal' and explained its function in the figure.

Very helpful!

Figure 2d:

☐ Combining ant-His moAb with a fluorescence-labeled secondary reagent in conjunction with a fluorescence-labeled streptavidin followed by two color imaging would enable the simultaneous detection of both protein and DNA on the same blot. The subsequent overlay of the fluorescence colors would directly reveal co-migration of protein and DNA.

This is a good idea, however, as we did not have access to an imager that can detect fluorescence different from SYBR green, we decided for stripping the western blot and reblotting with the streptavidin-HRP probe (Figure 2e). As both the band in the control lane and the signal in the lane with the DNase treated sample disappeared, we are confident that we did not have any signal carry-over from the initial western-blot. In both cases the band corresponding to the crosslinked NF1 protein was detected at the same height.

The alternative approach chosen by the authors is valid as well!

General: did the authors consider other search tools like the Xi search engine (recent work from the Rappsilber lab) or TagGraph (Nature Biotechnol. 37, 469-479)?

We did not consider these two search engines for this study, but we will look into them in future.

OK!

Reviewer #2 (Remarks to the Author):

The authors have provided all the clarifications we requested in this revised manuscript. As such, we think in its current form, it is suitable for publication in Nat Comm.

-Yamini Dalal and Aleksandra Nita-Lazar

NIH

Reviewer #3 (Remarks to the Author):

The authors have submitted a much-improved manuscript, primarily through the provision of an extensive number of additional controls and greater precision in the annotation of their spectra (reducing most of the speculation in the first draft). I believe there are no substantive issues preventing publication.

Point-to-point response to the comments of Reviewer #1

Maybe it escaped my notice but I cannot find a short but comprehensive description of the concept 'xlinked species' in the revised manuscript? It would be nice to add this to the final version!

We have now added an explanation of x-linked species to the legend of Figure 2e.

Please reconsider "Fig. 1C: RNPXL and manual annotation" The MS spectrum is not really readable (in a print out). Maybe a cartoon (hypothetical spectrum) emphasizing the detection of peptide ion fragments and peptide-nucleobase fragments as well as nucleobase ions would do a better job?

We changed the spectrum in Figure 1C and made it more readable.

I agree with the authors that these technicalities (absence of a chemical propionylation step prior to trypsin digestion) could provide a good explanation for the missed out crosslinks. But other scenarios might also come into play here. Some of these tryptic peptides are indeed too short in order to be captured by C18 in general. Others (peptides with a length of five to six amino acids) are often excluded by standard proteomic data processing approaches or are not retained by C18 STAGE tips (short and very hydrophilic ones). Therefore, I would like to recommend that the authors have a look into their data with the goal to determine which of the non-crosslinked peptides are already missing (escaping their detection).

Following the suggestions of reviewer #1, we analyzed all four histone proteins for theoretical peptides (Table R1). We identified theoretical peptides of five amino acid length only as part of a longer peptides with missed cleavage. Peptides with six amino acid length were present in both non-crosslinked and DNA crosslinked form. While we cannot exclude a C18 bias towards longer peptides, we do not believe that this majorly impacted our experimental workflow. In addition, smaller peptides are often missed out by search engines due to their smaller identification scores. We therefore believe that the overall gain of implementing a desalting method which is better suited for smaller peptides would be rather small.

Table R1: Theoretical tryptic peptides of human histone proteins. Peptides of 5 or 6 amino acids length are highlighted in red.

	Not identified	Identified non-crosslinked	Identified crosslinked
H2A - VGAGAPVYLAADVLEYLTAEL LELAGNAAR	X		
H2A - VTIAQGGVLPNIQAVLLPK		X	
H2A - AGLQFPVGR		X	
H2A - NDEELNK			X
H2A - HLQLAIR			X
H2A - TESHK	X		
H2A - GNYSER	X		
H2B - AMGIMNSFVNDIFER		X	
H2B - QVHPDTGISSK			X ¹
H2B - ESYSVVYK		X	
H2B - LLLPGELAK		X	
H2B - HAVSEGTK			X ¹
H2B - EIQTAVR	X		
H2B - LAHYNK		X	
H2B - IAGEASR	X		
H2B - STISR	X		
H2B - SAPAPK		X ²	
H2B - YTSAK	X		
H2B - PEPK		X ²	

H3 - FQSSAVMALQEACEAYLVGL FEDTNLCAIHAK	X		
H3 - YRPGTVALR	X		
H3 - EIAQDFK			X
H3 - STELLIR		X	
H3 - SAPATGGVK			X
H3 - DIQLAR			X ³
H3 - VTIMPK			X ³
H3 - LPFQR	X		
H3 - QLATK	X		
H4 - DNIQGITKPAIR			X
H4 - TVTAMDVVYALK		X	
H4 - ISGLIYEETR		X	
H4 - DAVTYTEHAK		X	
H4 - VFLENVIR		X	
H4 - TLYGFGG	X		

- 1) Identified with one missed cleavage
- 2) Identified as one unmodified peptide by one missed cleavage
- 3) Identified as one crosslinked peptide by one missed cleavage

Figure 4g: Again y12 is equivalent to the peptide molecular ion (in this spectrum the doubly charged species) that had lost C. The proposed C⁻ ion exhibits indeed strong ion abundance. Nevertheless, in the RNA field, nucleobase ions are rarely observed in case of mononucleotide crosslinks, because the nucleobase is believed to be covalently bonded to an amino acid side chain. Hence, observation of this ion will only be feasible in case the bond could dissociate during HCD (e.g. an amide bond crosslink between the amino group of C and the aspartate side chain).

While the free nucleobase could indicate a crosslink via the deoxyribose part, the existence of a nucleobase crosslink on the y3 ion clearly shows that the crosslink is nucleobase directed. Hence, we believe that the free nucleobase must indeed be dissociated from the crosslinked peptide during HCD fragmentation.

Yes, I would also BELIEVE so. Nevertheless, it should be made clear that this is a hypothesis.

We added a more detailed explanation of this hypothesis to the main text.

I am able to recapitulate the calculation of the authors (CO: 27.99491463 amu) resulting in a value of m/z 89.06026 for the proposed ion species. Radical based (induced by UV/oxygen) activation of the deoxyribose C5 position has been described in the context of DNA damages. In addition, the loss of cytosine or nucleobases in general (dissociation of the N-glycosidic bond) are well documented as well. But the molecular pathway resulting in the formal adduct formation equal to the monoisotopic mass of CO remains enigmatic. In consequence, Reim et al. are again just proposing another interesting hypothesis. Moreover, the authors readily admit that there exists a large distance between L178/K181 and the C in the crystal structure. The latter might indicate structural flexibility. Nevertheless and in light of the fact that the manuscript should mainly provide proof-of-principle experimental results, I strongly recommend to move Fig. 5e, f and g to the supplementary section. Likewise, it would make sense to remove the L178/K181 crosslink from the current figure 5f. In my point of view these results are not essential for convincing the general reader about the novelty and technological relevance of the study!

We followed the recommendations of reviewer #1 and moved the Fig. 5e-g to Supplemental Figure 4.

Supplementary Figure S3: there is one peptide ion (from HES-2, KPLLEL + [TT], MH33+) that exhibits a Δm of 10.7 ppm even though the methods section claims that the DB search was performed with a maximum allowed mass deviation of +/- 10 ppm.

This peptide was identified by RNP-XL as an AA-H2O crosslink. However, as the spectra contained a marker ion for thymine, and the mass of the AA-H2O adduct is very close to TT (626.1152 vs. 626.1027), we manually annotated the peptide as a TT crosslink, even if the mass deviation slightly exceeded the 10 ppm limit.

The mass deviation between the RNP-XL proposed AA-H2O crosslink and a TT adduct is substantial (almost 20 ppm)! What was the (measured) mass deviation for the proposed AA-H2O crosslinked species?

The Δm of a hypothetical AA-H2O crosslinked species is 1.6 ppm. Given the presence of two thymine-shifted fragments and the thymine marker ion in the spectra, our data indicates that the TT crosslink has a higher probability despite the higher mass error.

Generally speaking, it would be important to know if the MS1 spectra were recalibrated at the global level (as performed for instance by MaxQuant)? Since on one hand the PD version is not indicated in the methods section and on the other hand I am not familiar with the details of PD data processing this is an important point because many spectra do actually show a Δm of greater 5 ppm, which at least in my experience is unusual for correct IDs on data acquired on a HF-type orbitrap instrument, except recalibration was not carried out. In this context I would also like the *authors* to provide the Δm values for the major fragment ions in the ion spectrum tables requested in my comments above (Figure 4). Information on the latter will provide additional trust in the manual curation of MS/MS spectra.

To our knowledge and based on the documentation, the RNP-XL workflow does not contain a recalibration step. We have followed the suggestions of reviewer #1 and included Δm values in Supplemental table 1 containing all theoretical and identified fragment ions.

This is very nice. In general, most of the fragment ions have a very low mass deviation. Still, quite some of the precursor/parent ion masses exhibit Δm values of greater 5 ppm. Apart from the absence of a recalibration step (in the workflow) this is maybe also a problem related to Proteome Discoverer software underperforming in accurate m/z determination from DNA-peptide heteroconjugates (contain a relative higher number of the 'mass deficient' oxygen atom as compared to amino acids/peptides)? In other words do the authors observe a better fit to the theoretical mass for "normal" tryptic peptides?

Following the suggestions of reviewer #1, we analyzed the median delta mass for non-crosslinked and crosslinked peptides in each experiment. In fact, we identified a better mass fit for non-crosslinked peptides.

	Median Δm non-xl	Median Δm xl
TBP	2.6	3.4
NF1	2.4	6.3
Nucleosome	1.0	5.4
ESC	2.0	3.65